# Lateral membrane organization as target of an antimicrobial peptidomimetic compound

Adéla Melcrová ®[1], Sourav Maity[1], Josef Melcr[2], Niels A. W. de Kok ®[3], Mariella Gabler[1], Jonne van der Eyden[1], Wenche Stensen ®[4], John S. M. Svendsen ®[4], Arnold J. M. Driessen ®[3], Siewert J. Marrink ®[1,2] & Wouter H. Roos ®[1] ✉

Antimicrobial resistance is one of the leading concerns in medical care. Here we study the mechanism of action of an antimicrobial cationic tripeptide, AMC-109, by combining high speed-atomic force microscopy, molecular dynamics, fluorescence assays, and lipidomic analysis. We show that AMC-109 activity on negatively charged membranes derived from *Staphylococcus aureus* consists of two crucial steps. First, AMC-109 self-assembles into stable aggregates consisting of a hydrophobic core and a cationic surface, with specificity for negatively charged membranes. Second, upon incorporation into the membrane, individual peptides insert into the outer monolayer, affecting lateral membrane organization and dissolving membrane nanodomains, without forming pores. We propose that membrane domain dissolution triggered by AMC-109 may affect crucial functions such as protein sorting and cell wall synthesis. Our results indicate that the AMC-109 mode of action resembles that of the disinfectant benzalkonium chloride (BAK), but with enhanced selectivity for bacterial membranes.

Ever since the introduction of antibiotics, bacteria have developed resistance against these compounds. For instance, methicillin-resistant variants of the Gram-positive bacterium *Staphylococcus aureus* (MRSA) appeared already several decades ago[1]. This discovery spurred the search for new types of antibiotics, however, this quest has met only limited success. In fact, antimicrobial resistance was declared by the World Health Organization as one of the top 10 global threats in medical care, with *S. aureus* as one of the leading pathogens[2]. Natural antimicrobial peptides and their mimics are promising scaffolds for the development of new antibiotics. Antimicrobial peptides are an integral part of our immune system, and as such they have potent antimicrobial activity against a broad range of bacteria[3–5]. However, they have several drawbacks, such as high production costs, cytotoxicity to red blood cells, and development of bacterial resistance[6,7]. One way to overcome the antimicrobial resistance is to stabilize the

antimicrobial peptides by using non-native amino acids or modified peptide chains generating non-natural peptide mimics − peptidomimetics − which are stable in vivo, not harmful to human cells, and not suffering from resistance[5,7–10]. Here we focus on a promising membrane-active peptidomimetic AMC-109 (Fig. 1a)[11], a cationic artificial tripeptide with capped C-terminus that is relatively simple to synthesize, has a broad antimicrobial activity, and can be processed in large-scale industry[7]. AMC-109 follows the minimal pharmacophore model developed in the lab of Svendsen[12] stating a simple rule that a minimum of four amino acid residues are required for significant antibacterial activity − two of them cationic, the other two hydrophobic[7,11,12]. In AMC-109 the two hydrophobic residues are replaced by a single artificial residue encompassing the bulk and hydrophobicity of two individual tryptophans. The presence of three tert-butyl groups in the 2-, 5- and 7-positions on the indole ring makes

[1]Molecular Biophysics, Zernike Institute for Advanced Materials, Rijksuniversiteit Groningen, Groningen, the Netherlands. [2]Molecular Dynamics, Groningen Biomolecular Sciences & Biotechnology Institute, Rijksuniversiteit Groningen, Groningen, the Netherlands. [3]Molecular Microbiology, Groningen Biomolecular Sciences & Biotechnology Institute, Rijksuniversiteit Groningen, Groningen, the Netherlands. [4]Department of Chemistry, UiT Arctic University of Norway, Tromsø, Norway. ✉e-mail: w.h.roos@rug.nl

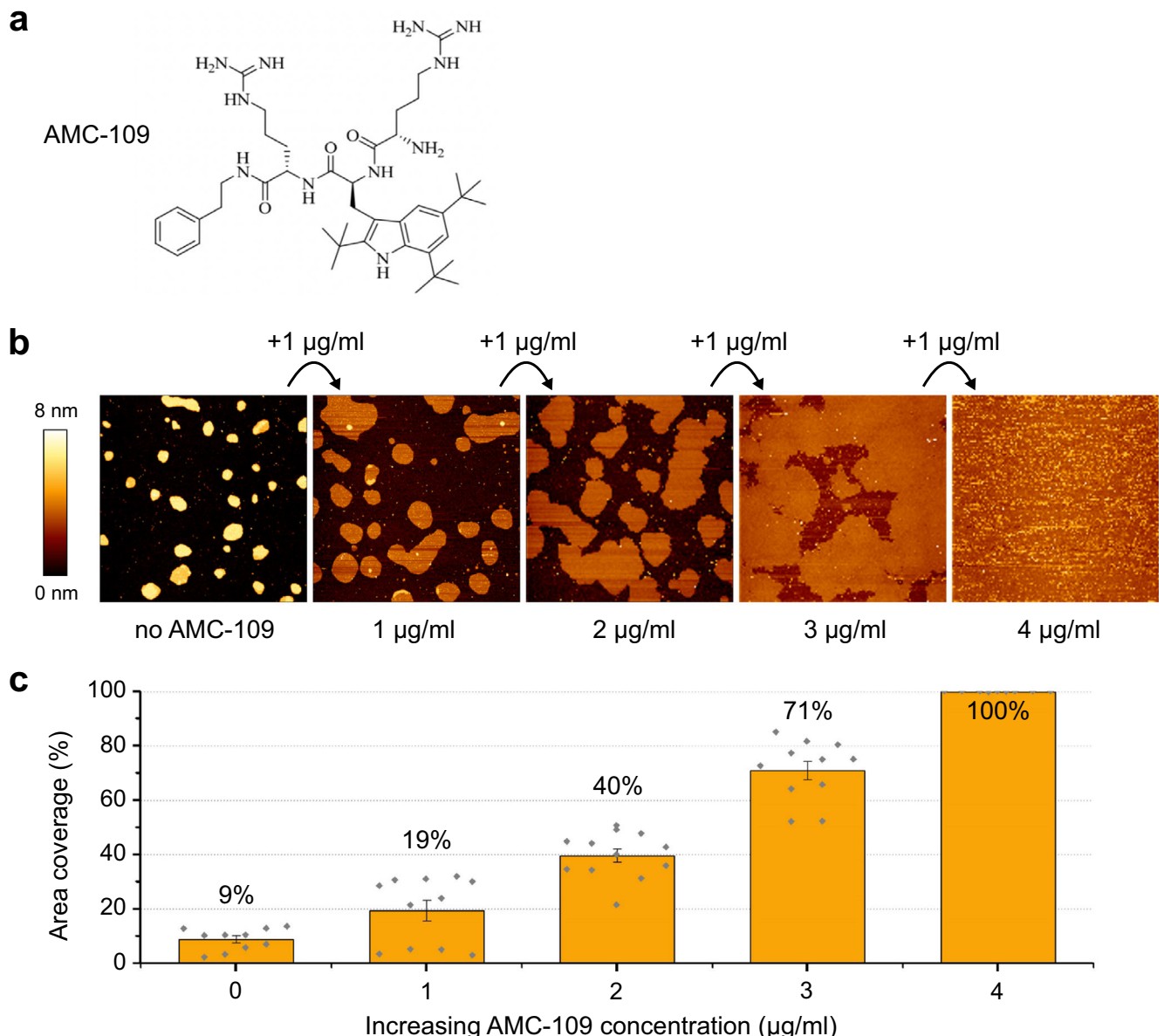

**Fig. 1 | AFM imaging experiments reveal the effect of AMC-109 addition to *S. aureus* lipid membranes. a** AMC-109 chemical structure. **b** *S. aureus* lipid membranes upon treatment with AMC-109. Membranes are visualized untreated (left) and treated by subsequent additions of 1 µg/ml AMC-109 (0, 1, 2, 3, 4 µg/ml). Changes in the background colour originate from a combination of the membrane thinning and coverage of the mica by AMC-109. Representative of 3 independent experiments. **c** Area of the mica surface covered by the membrane upon subsequent additions of 1 µg/ml AMC-109. For each concentration, coverage of 10–12 areas in a 3×3 µm image size were evaluated. 3×3 µm images for area evaluation were gathered across 3 independent experiments (3–4 images for each AMC-109 concentration per experiment). Error bars represent standard error of the mean. Individual data points are depicted in grey. Source data for panel **c** are provided as a Source Data file.

the side chain of this amino acid 2.5 times larger and more lipophilic than the indole of tryptophan. A regular tryptophan variant of AMC-109 did not show good antimicrobial activity[13]. The modified tryptophan residue is, hence, a key element of AMC-109, as it allows the peptide to obtain high activity with only three amino acids. The positive charge in AMC-109 is provided by the two arginine residues.

The antimicrobial activity of AMC-109 was positively tested against two Gram-negative bacteria, *Escherichia coli* and *Pseudomonas aeruginosa*, as well as a multitude of Gram-positive *S. aureus* strains, including the strains resistant to common antibiotics such as methicillin or vancomycin[11,14]. AMC-109 also proved a low cytotoxicity in these studies showing a ~100-fold difference between the minimal inhibitory concentration (MIC) against *S. aureus* strains (<2 µg/ml) versus erythrocyte toxicity (175 µg/ml)[11]. Clinical trials bring promising results so far. In nasal decolonization studies conducted in Sweden, patients infected with two variants of *S. aureus* were treated for three days with AMC-109, which significantly reduced infection[15]. Integrating of AMC-109 into wound dressings and gels leads to significantly higher antibacterial activity compared to current clinical standard therapy[16]. Moreover, AMC-109 was demonstrated to have more than five times longer post-antibiotic effect against various *S. aureus* strains than commonly used mupirocin, and could hence be applied less frequently or in smaller doses to maintain its antimicrobial activity[17]. The promising pharmaceutical studies raise the question what the molecular mechanism of the antimicrobial activity of AMC-109 is. Detailed understanding of the antibiotic effects at the molecular level will support identifying new drug targets in resistant bacteria, and can be used in further development of new potent antimicrobial drugs.

**Table 1 | Lipidomic analysis of total lipid extracts from S. aureus showing the relative densitometric response of the three major glycerophospholipid classes (molybdenum blue) and two glycolipid classes (α-naphtol) on thin layer chromatography**

| Glycerophospholipids | | |
|---|---|---|
| Compound[a] | Charge at pH ~ 7 | Relative densitometric response (Molybdenum blue) |
| CL | −2 | 9 % |
| PG | −1 | 66 % |
| Lysyl-PG | +1 | 24 % |
| Glycolipids | | |
| Compound[a] | Charge at pH ~ 7 | Relative densitometric response (α-naphtol) |
| 1-hex-DAG | 0 | 15 % |
| 2-hex-DAG | 0 | 85 % |

Full data are shown in Fig. S1.

[a]CL, cardiolipin; PG, phosphatidylglycerol; Lysyl-PG, lysyl-phosphatidylglycerol; 1-hex-DAG, monoglycosyldiacylglycerol; 2-hex-DAG, diglycosyldiacylglycerol.

Here we test how AMC-109 affects the membranes of S. aureus on a molecular/nanoscopic level by a combination of atomic force microscopy (AFM), high-speed AFM (HS-AFM), fluorescence leakage assays, molecular dynamics (MD) simulations, mass spectrometry-based lipidomic analysis and gel chromatography, yielding a comprehensive view on the complex molecular mechanism of AMC-109 activity on the membrane of S. aureus.

## Results and discussion

### Lipid extracts as a model of S. aureus cytoplasmic membrane

In order to scrutinize the interactions of AMC-109 with the membrane, we extracted lipids from S. aureus cells[18] to be used as bacterial model membranes for AFM and HS-AFM investigations, techniques that have already been shown insightful for antibiotics mechanism studies[19,20]. The lipids were characterized through lipidomic analyses (Table 1, Fig. S1). The lipid part of the S. aureus membrane is a complex mixture of mainly glycerophospholipids, followed by glycolipids[21–23]. Mass spectrometry and thin-layer chromatography confirmed the presence of both these lipid classes but couldn't measure their relative ratio in the membrane. We thus analyse these two major lipid classes present in the total lipid extracts from S. aureus separately (Table 1). We demonstrate that the most abundant glycerophospholipid is the negatively charged phosphatidylglycerol (PG, ~66% of all glycerophospholipids) and the most abundant glycolipid is diglycosyldiacylglycerol (~85% of all glycolipids). Overall, the lipidomic analysis confirms the presence of the negatively charged molecules in the phospholipid membranes of S. aureus, which results in the high attractivity for the cationic AMC-109. The analysis of the acyl chains distribution in the phospholipids fraction showed that the majority of the phospholipids contains saturated acyl chains, often with iso- or anteiso-branching complying with previous S. aureus lipidomic studies[21,24]. Moreover, the lipid extracts also contain a significant portion of unsaturated lipids (~10% of PG lipids, Fig. S1f), which is typically not reported for S. aureus[22,25].

### AMC-109 changes structural properties of S. aureus lipid membranes

AFM imaging was employed to observe the effects of AMC-109 on supported membranes prepared from the S. aureus lipid extracts. The AFM images at increasing concentrations of AMC-109 (Fig. 1b) reveal effects on the membranes already at concentrations of 1–2 μg/ml. Interestingly, this concentration coincides well with the reported MIC that induces antimicrobial activity on living S. aureus both wild-type[11] and a large number of resistant S. aureus clinical isolates, including MRSA (96 strains), vancomycin-intermediate S. aureus (VISA, 33 strains), and vancomycin-resistant S. aureus (VRSA, 13 strains)[14]. Upon exposure of the S. aureus lipid membranes to AMC-109 the membrane expands (Fig. 1b). For concentrations up to 2 μg/ml, this is a relatively fast process reaching an equilibrium state after several minutes (Fig. S2a). The expansion at 3 μg/ml is more gradual, slightly changing even after 40 min (Fig. S2b). The relative surface coverage starts around 10% without AMC-109 and gradually increases to reach 100% at 4 μg/ml of the peptidomimetic (Fig. 1b, c). The observed membrane expansion will lead to lateral stress in confined areas. In living bacteria, it is expected that expansion will be less pronounced due to the confinement of the membrane. The consequent built-up in the stress could eventually be partially released by an increased roughness of the membrane surface or the formation of membrane protrusions.

### AMC-109 clusters and dissolves lateral lipid domains in S. aureus lipid membranes

AFM imaging reveals a co-existence of regions with higher and lower thickness in the S. aureus lipid membranes (Fig. S3). However, these regions were typically smeared out suggesting a dynamic nature of these regions. We, hence, moved to HS-AFM to improve the temporal resolution[26,27]. HS-AFM reveals that the higher thickness regions in the untreated S. aureus lipid membranes are highly mobile lateral lipid domains (Fig. 2a, Supplementary Movie 1) that maintain roughly circular shape with a strikingly narrow size distribution of ~30 nm in diameter (30.9 ± 0.4 nm, $N = 269$, Fig. S4).

The observed phase separation probably arises from the complex composition of the lipid extracts, which apart from phospholipids also contain glycolipids and a portion of cardiolipin[28–31]. The domains display a rapid lateral movement inside the untreated membrane (Fig. 2a, Supplementary Movie 1). After the addition of AMC-109 at the concentrations above the MIC (Fig. 2b, Supplementary Movie 2), however, the domains accumulate together (Fig. 2b 1–2 min) before they gradually disappear (Fig. 2b 3–4 min). The accumulation of the domains occurs within tens of seconds after the addition of AMC-109 to the membranes. The accumulated domains gradually dissolve after about 3–5 min and only then the membrane starts expanding and significantly thinning (Fig. 2b, c, Supplementary Movie 2). Interestingly, lower AMC-109 concentrations (around 1 μg/ml) are enough to induce the accumulation and domain dissolution, which significantly alters the mechanical properties of the membrane, increasing its Young's modulus (Fig. S5, Supplementary Note 1). These concentrations, however, do not lead to full membrane expansion, which we observe only at higher AMC-109 concentrations. We measured the height of the S. aureus lipid membrane over time after adding 4 μg/ml AMC-109. The time evolution of the membrane height $h$ is displayed in Fig. 2c. A substantial thinning of ~1 nm of the continuous part of the membrane, i.e. the part not including the domains, is observed. In these experiments it is assured that the mica background does not get covered with AMC-109, allowing for accurate height measurements (Fig. S6, S7).

The disc-shaped lateral domains in our S. aureus lipid membranes (Fig. 2a) are reminiscent of native functional membrane microdomains (FMM) from living bacteria[32]. FMMs in the cytoplasmic membrane of S. aureus fulfill important functions such as peptidoglycan synthesis, membrane lipid metabolism, membrane transport, protein quality control, virulence, and also oligomerization of low-affinity penicillin-binding protein (PBP2a) responsible for the resistance against penicillin, methicillin and other β-lactam antibiotics[33–37]. It was shown that FMMs contain cardiolipins and glycolipids[31,37], which are also present in our S. aureus lipid membranes. Dissolution of FMMs in living bacteria leads to their death as the bacteria lose the multiple functions associated with these domains. Moreover, MRSA bacteria without the

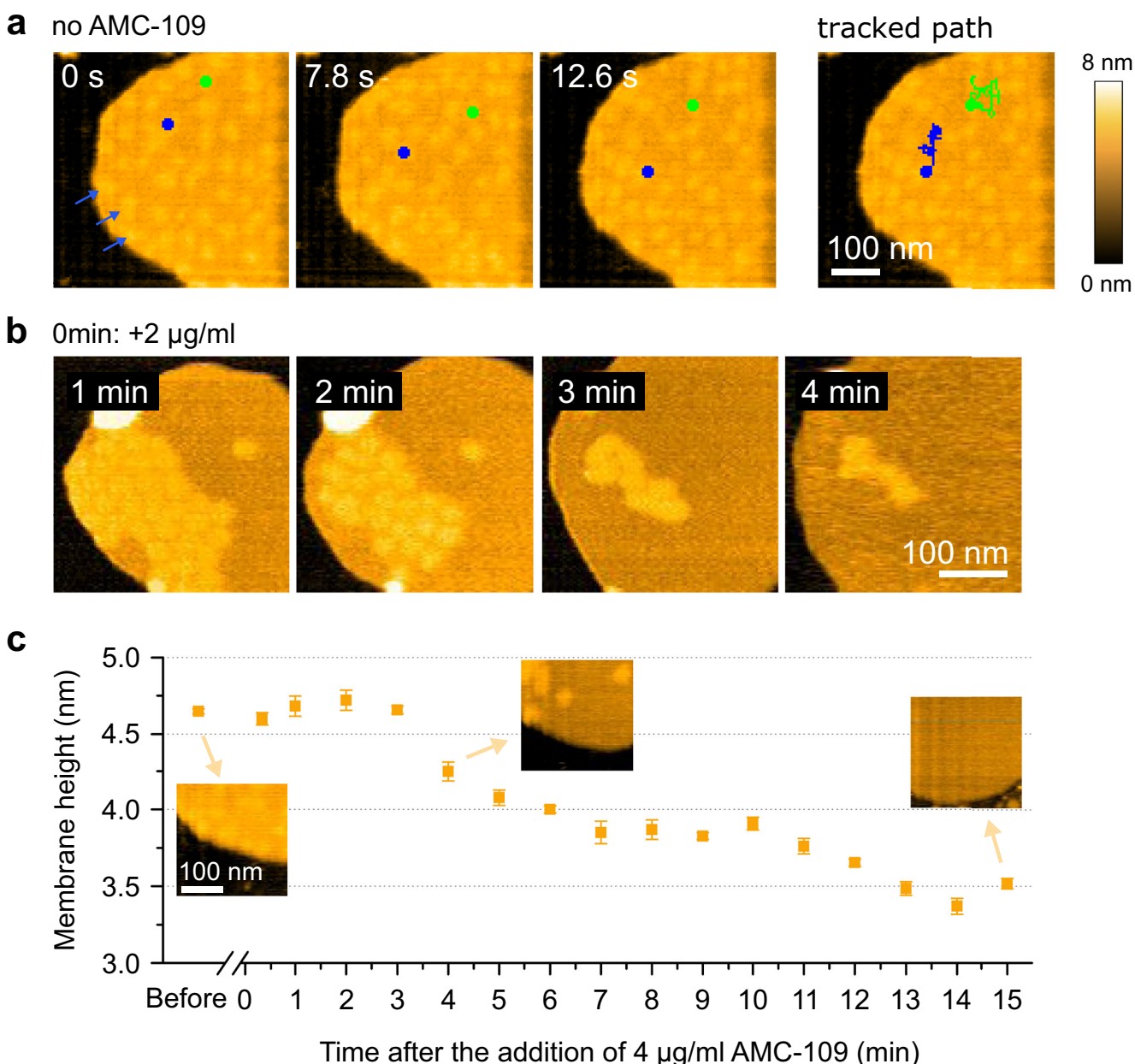

**Fig. 2 | HS-AFM imaging of the AMC-109 effects on *S. aureus* lipid membranes supported on mica. a** HS-AFM images of the untreated membrane showing the diffusion dynamics of lateral lipid domains (blue arrows). Dynamic path of two lateral domains (blue and green discs) were tracked in the 0.3 s per frame movie. Nanodomains diffusion was observed in 18 independent experiments. **b** HS-AFM images of the membrane in time after the addition of 2 µg/ml AMC-109. For an example of a 0 min frame see panel **a**. Lateral domains accumulate together, then gradually dissolve. This is followed by expansion and thinning of the membrane. Similar behavior was observed in 5 independent experiments. **c** Membrane thickness evolution over time after the addition of 4 µg/ml AMC-109. Measured from HS-AFM images at the continuous parts of the *S. aureus* lipid membrane surrounding the lateral lipid domains. Mean membrane height and standard errors of the mean are displayed. Data points are each the mean of N > 20 individual measurements (N = 363, 49, 22, 21, 74, 31, 52, 93, 37, 41, 71, 27, 35, 34, 27, 24, and 45, from left to right respectively. N represents number of point height measurements from the HS-AFM image at the specified time. Each height measurement is given as a difference between the height on top of the membrane and the height of the background.). Insets shows membrane details before the AMC-109 addition, and at t = 4 min and t = 15 min after. Average height before the AMC-109 addition is 4.65 ± 0.02 nm. At time 13–15 min the height settles around 3.5 nm. Source data for panel **c** are provided as a Source Data file.

FMMs are no longer resistant to β-lactam antibiotics, including methicillin[33,34,37]. We, hence, suggest that the AMC-109 treatment could be further enhanced by the use of β-lactam antibiotics.

## Molecular picture of AMC-109 attacking model membranes

We have employed MD simulations using the coarse-grained Martini 3 force field[38,39] to obtain a molecular picture of AMC-109 activity at model lipid membranes. Fig. 3a shows the chemical structure of the

AMC-109 molecule together with its Martini 3 representation and dimensions. The amphiphilic nature of AMC-109 molecules drives their fast spontaneous self-assembly into micellar aggregates within the first 100 ns of equilibration simulations[40]. Such spontaneously formed aggregates adopt a globular or slightly elongated shape with a characteristic size of 4 nm in diameter (Fig. 3b).

From both AFM and HS-AFM imaging, we see that such aggregates also attach to the mica surface. Fig. 3c (Supplementary Movie 3) shows

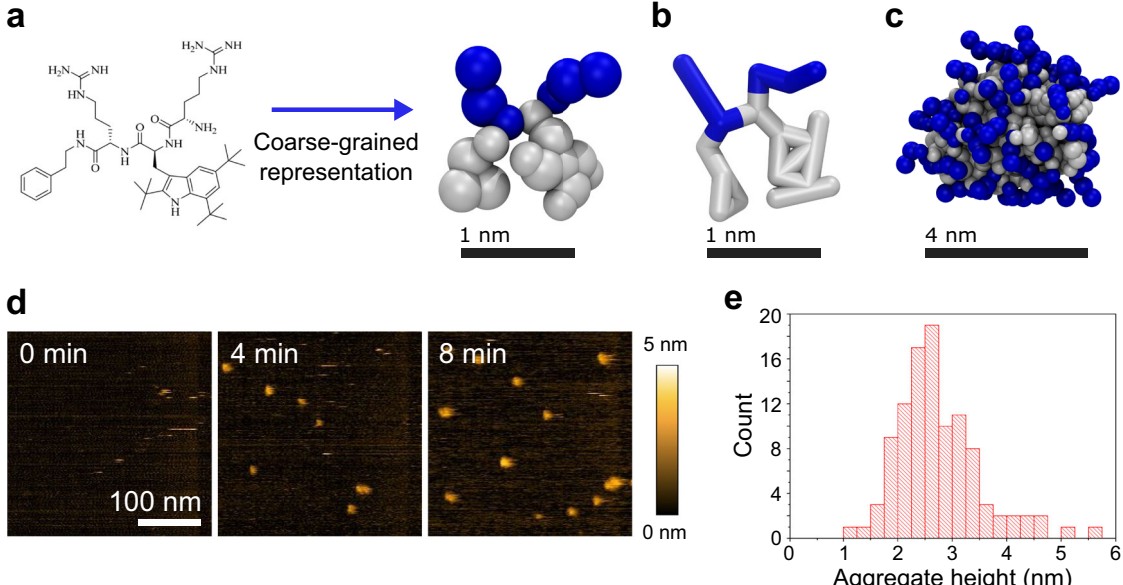

**Fig. 3 | Aggregation properties of AMC-109. a** AMC-109 chemical structure and its representation in Martini 3 coarse grained model. Cationic residues are depicted in blue, artificial hydrophobic residues mimicking phenyl-alanine (small triangle) and tryptophan with three tert-butyl groups (large planar structure) are depicted in grey. **b** Alternative representation of the AMC-109 molecule showing the connections between the interaction sites in the model. **c** Representative self-assembled aggregate of AMC-109 molecules formed after ∼0.5 μs of the MD simulation. **d** HS-AFM images of micellar aggregates adsorbed on mica after the addition of 5 μg/ml AMC-109. A representative of 2 independent experiments is shown. **e** Histogram of the measured aggregate height. Average aggregate height was evaluated as 2.75 ± 0.08 nm (N = 104). Source data for panel **e** are provided as a Source Data file.

how after adding 5 μg/ml AMC-109 to the buffer solution, individual ball-shaped aggregates appear on the mica. The height of these aggregates attached on a solid surface is 2.75 ± 0.08 nm (N = 104, Fig. 3d) being in line with the size of a single aggregate in the bulk predicted by MD simulations (Fig. 3b). After the attachment of the first aggregate on mica during our HS-AFM experiment, the adsorption process continues and the aggregates eventually form a continuous layer of molecules with a thickness of 1–3 nm (Supplementary Movie 3).

Bacterial membranes possess a net negative charge (∼70% negatively charged lipids in *S. aureus* lipid membranes[41]), while the outer leaflet of eukaryotic membranes is neutral[42,43]. To demonstrate the selectivity of AMC-109 to bacterial membranes, we have varied the surface charge of the simulated model membranes. The result is series of MD simulations with mixed POPC/POPG bilayers (POPG content 0–100 mol%; simulation setup see Fig. S8.) In the simulation setup, we start with AMC-109 monomers in the bulk water surrounding the membrane. In the beginning, the AMC-109 molecules assemble into the aggregates but there is a minute portion of AMC-109 monomers that encounter the membrane before they have a chance to be a part of an aggregate. Such monomers incorporate into the hydrophobic core of the membrane regardless its charge. They are responsible for the non-zero adsorbed amount of AMC-109 molecules in the case of pure POPC membrane (Fig. 4a, 0–10% negatively charged lipids), and are stable in the bilayer for the rest of the simulation (Fig. S9).

The adsorption curves in Fig. 4 of the AMC-109 aggregates demonstrate their increasing attractivity to membranes with a higher negative charge. Notably, the adsorption of AMC-109 always saturates close to the point when the net charge of the membrane is neutralized. (Membrane saturation with AMC-109 aggregates can be viewed at Supplementary Movie 4.) After the membrane is neutralized, it is no longer a preferred energy minimum location for AMC-109. The rest of the AMC-109 molecules then stay in solution in the form of aggregates. In particular, membranes with 20 mol% POPG or more adsorb on average one AMC-109 molecule (charge +3) per three molecules of POPG (charge −1), while membranes with small

negative charge (≤10 mol% POPG) attract negligible amounts of AMC-109 molecules.

The insertion of a single AMC-109 aggregate is depicted in Fig. 4b (and Supplementary Movie 5). First, the aggregate makes direct contact with the negatively charged lipid headgroups in the membrane (red in Fig. 4b), then it remodels the upper leaflet of the attacked membrane pushing the lipids from each other and inserting the AMC-109 molecules from the whole aggregate in between the phospholipids. Interestingly, the AMC-109 molecules from a single aggregate enter only the upper membrane leaflet and do not translocate into the opposite one within the simulation time scales. Once inserted into the hydrophobic interior of the membrane, AMC-109 molecules move as monomers, and do not further cooperate towards any major membrane disruption. The whole process of the aggregate adsorption and dissolution into the membrane takes ∼300 ns in the simulation. The time needed for the saturation of the membrane with the AMC-109 molecules is ∼2 μs for membranes with higher content of the negative charge (>30% POPG, Fig. 4a), and longer (>5 μs) for membranes less attractive for AMC-109 (≤30% POPG). As a consequence of the AMC-109 insertion into the membrane, it becomes thinner and expands laterally leading to an overall more disordered membrane (Fig. S10).

We further test the selectivity of the aggregates towards negatively charged membranes by HS-AFM experiments (Fig. 4c, d), where we observe whether the expansion of synthetic lipid membranes occurs in the first 10 min after exposure to 4 μg/ml AMC-109. Neutral (POPC) and positively charged membranes (DOTAP/POPC, 60/40 mol %) increase their surface area by 16 ± 5% and 49 ± 3%, likely due to the incorporation of a few AMC-109 monomers from the solution. The negatively charged (POPG/POPC 60/40 mol%) grows by 880 ± 190%. The significantly higher effects to the negatively charged membranes can be explained by the additional incorporation of the micellar aggregates, as predicted by the MD simulations. The large area increase of the membrane upon AMC-109 binding, as also reported in Fig. 1, results partially from the insertion of large amounts of AMC-109 into the phospholipid bilayer. For bacterial membranes up to 25 AMC-109 molecules per 100 lipids can be inserted, as the simulation results

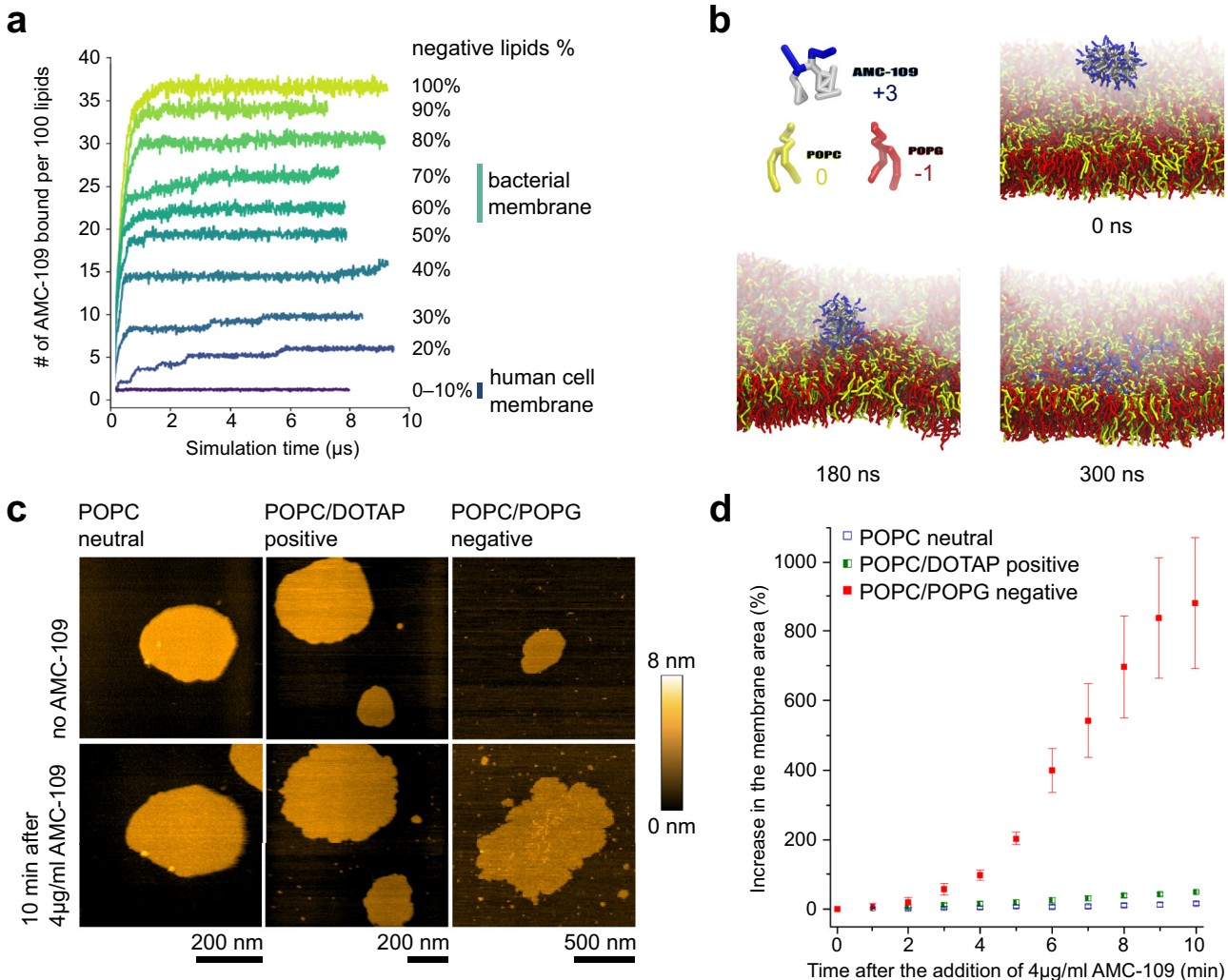

**Fig. 4 | Interaction of AMC-109 aggregates with model membranes. a** MD results on the number of AMC-109 molecules attached to the membrane per 100 lipids. An increase in a curve corresponds to the attachment of one or more aggregates into the membrane. The curves for 0 and 10% of negatively charged lipids both show no attachment of the AMC-109 aggregates. Levels of the negative charge corresponding to the membranes of human and bacterial cells are highlighted. **b** Interaction of a single AMC-109 aggregate with a symmetric POPG/POPC (60/40 mol%) membrane. The aggregate attaches to the membrane and gradually all the AMC-109 monomers dissolve in between the lipids. The simulation was run with multiple AMC-109 aggregates (Fig. S8). Other AMC-109 aggregates and water are hidden for clarity. Time stamps refer to the time of the simulated MD trajectory. **c** Representative HS-AFM images of the synthetic lipid membranes before and

10 min after the addition of 4 μg/ml AMC-109. Experiments were performed on neutral POPC, positively charged DOTAP/POPC (60/40 mol%), and negatively charged POPG/POPC (60/40 mol%). Growth of supported membranes deposited on mica was monitored in time after the addition of the AMC-109. **d** Quantitative analysis of the increase in the surface area of synthetic lipid membranes in time after treatment with 4 μg/ml AMC-109. Data correspond to representative images shown in panel **c**. Negatively charged membranes prove to be significantly more affected than neutral and positively charged ones. Averages and standard errors of the mean measured for each lipid composition for 3 individual experiments on 6–7 membrane patches are shown. Individual growth curves used for the displayed mean calculation are at Fig. S11. Source data for panels **a, d** are provided as a Source Data file.

in Fig. 4a reveal. Apart from this contribution, the membrane also gets thinner, which is reported by HS-AFM measurements in Fig. 2c. Membrane thinning will lead to lipid rearrangements and together with the increase in the material due to AMC-109 incorporation, this leads to the observed membrane spreading.

Taken together, the selectivity of AMC-109 to bacteria seems to arise from two main factors. First, the amphiphilic nature of AMC-109 drives the formation of nanometer-sized aggregates hiding their hydrophobic parts and exposing their cationic groups. MD simulations agree on the shape and characteristic dimensions of such aggregates with HS-AFM images (Fig. 3c–e), which also demonstrate their existence at concentrations around the MIC for *S. aureus*. Second, self-assembly of AMC-109 into aggregates leads to the minimization of the free energy. So unless the AMC-109 is attracted to the membrane by its negative charge, AMC-109 rather prefers staying in a hydrophobic

aggregate instead of entering the hydrophobic membrane[40]. This prevents interaction of AMC-109 with neutral membranes lowering the cytotoxicity of the compound for human cells[42,44]. Both MD simulations and HS-AFM experiments show that this barrier is lowered by the bacterial membrane surface charge, which also facilitates the attraction of the aggregates. This leads to their loading into the bacterial membrane resulting in severe damage of the membrane lateral organization, which we suggest to be the cornerstone of the antimicrobial activity of AMC-109. Potentially, this could even have implications for cancer treatment as multiple studies already investigated the possibility of therapeutic application of cationic peptides[39–41].

## AMC-109 does not form membrane pores
AMC-109 has been designed as a mimic of a membrane-active antimicrobial peptide[11]. Such peptides have been reported to disrupt

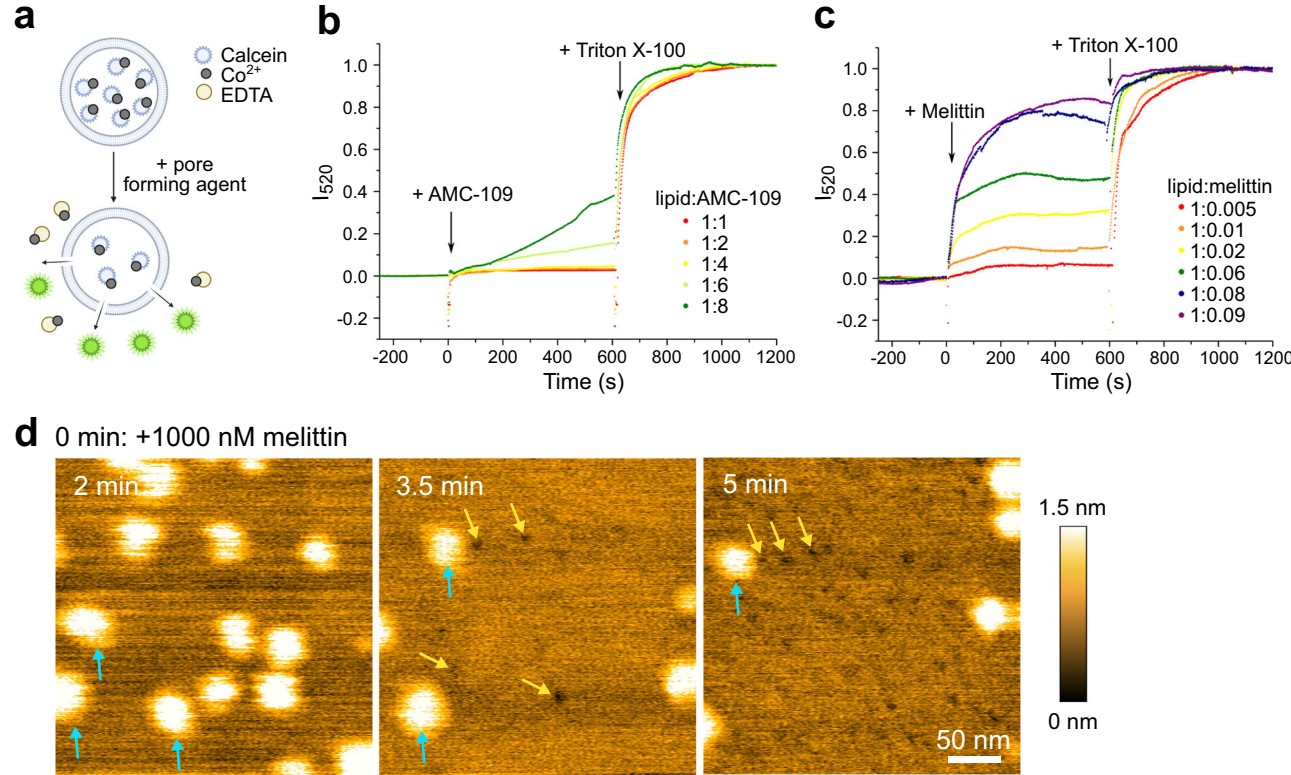

**Fig. 5 | Pore-forming activity of AMC-109 and melittin. a** Cobalt-calcein leakage assays principle. Non-fluorescent cobalt-calcein complex is trapped inside of the membrane liposomes. The addition of a perforating agent leads to their leakage out of the liposomes, where cobalt is extracted from the complex by EDTA resulting in an increased calcein fluorescence intensity. Schematics created in BioRender.com. **b,c** Calcein leakage induced by AMC-109 and melittin in POPG/POPC/POPE (60/30/10 mol%) liposomes. Arrows indicate the addition times of the antibiotic agent and detergent Triton X-100 to fully rupture the liposomes and obtain maximum fluorescence signal. 100 µg/ml liposomes were used for the experiments. **b** AMC-109 does not induce any leakage up to a ratio of 6 AMC-109 molecules per single lipid. **c** Melittin displays gradual leakage corresponding to its pore forming activity. **d** HS-AFM images of *S. aureus* lipid membranes after the addition of 1000 nM melittin. Blue arrows point at the lateral domains, visible as bright parts of the membrane. Yellow arrows point at the formed pores. Initially a lot of lateral domains are visible (left), pores start appearing after a few minutes of melittin exposure (middle), and gradually more pores are being formed (right). Representative of 3 independent experiments is shown. Source data for panels **b** and **c** are provided as a Source Data file.

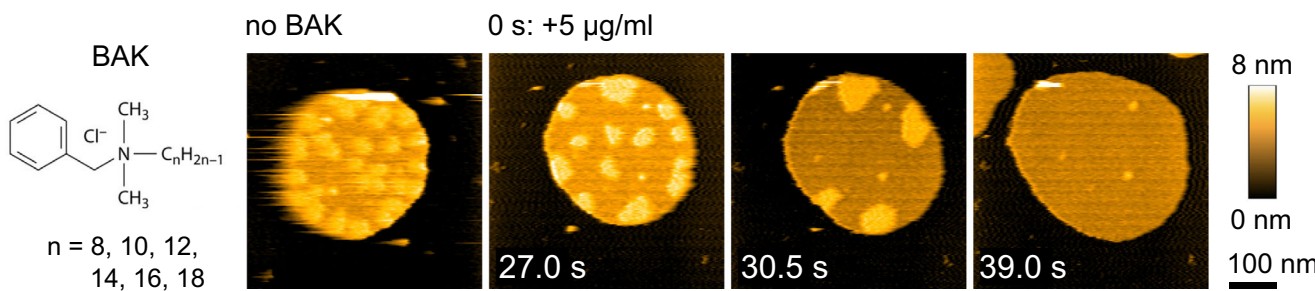

**Fig. 6 | Effects of the disinfectant benzalkonium chloride (BAK) on *S. aureus* lipid membranes.** Chemical structure of BAK and HS-AFM images of the membrane before and after the addition of 5 µg/ml BAK. The same effects were observed in 5 independent AFM or HS-AFM experiments.

membrane integrity by for instance, creating transmembrane pores or by disintegrating the membrane[45,46]. Our results on AMC-109 activity, however, suggest that AMC-109 incorporates into the bacterial membrane and changes the lateral lipid organization dissolving the lateral domains, which changes both dynamics of the lipids and the membrane material properties. Neither simulations, nor experiments reveal any porous defects in the membrane. To confirm that we did not overlook any small membrane perturbations, we performed cobalt-calcein leakage assays and compared the leakages induced by AMC-109 and by the well-known pore-forming peptide melittin[47–51] (Fig. 5). AMC-109 induces no leakage of calcein in POPG/POPC/POPE (60/30/10 mol%) liposomes up to very high concentrations of 6 AMC-109 molecules per single lipid (600 µg/ml AMC-109 added to 100 µg/ml

liposomes). Only at these very high concentrations of AMC-109, we observe slow but continuous leakage without any saturation behavior (Fig. 5b), which corresponds to the overall disruption of liposomes[52,53]. In contrast, pore-forming melittin induces pores in the liposomes yielding a gradual leakage corresponding to the creation of individual pores (Fig. 5c)[10,52,54].

The striking differences in membrane pore-forming activity of melittin and the lack of pore formation after AMC-109 treatment are visualized by HS-AFM (Fig. 5d). While AMC-109 largely affects lateral domain organization (Fig. 2) but does not induce detectable membrane pore formation (Figs. 1–2, and 5b), adding melittin leads to the formation of individual small pores (yellow arrows in Fig. 5d, Fig. S12). Such pores, with diameters below 10 nm, are distributed over the

whole membrane surface (Fig. 5d, Fig. S12), which is accordance with previous AFM studies[47,48]. Melittin also slows down the lateral movement of the lipid domains, and some of the domains disappear (Fig. 5d). In summary, these observations support our conclusion that AMC-109 does not perforate the *S. aureus* membrane in a manner comparable to pore-forming peptides but instead acts by affecting its lateral organization.

## AMC-109 acts like a bacteria-selective disinfectant

The mechanism of the activity of AMC-109 hence does not resemble that of pore-forming peptides, or of other membrane-active antibiotics, which perforate the membranes via various types of pores or carpet-like membrane coverage and disruption[3,45,55]. Incorporation of AMC-109 molecules in between the lipids of a single membrane leaflet and effects on lateral organization in fact resemble the activity of small disinfectant molecules like ethanol, benzalkonium chloride (BAK) or benzyl alcohol[56–66].

Ethanol (Fig. S13) was previously shown to thin model lipid membranes, lower the order of the lipid tails, and dissolve lateral cholesterol-rich domains in membranes[57–60]. BAKs are molecules containing a quaternary nitrogen associated with hydrophobic headgroup and a single hydrophobic acyl chain of a varying length (Fig. 6). These surface-active molecules are believed to insert in between the lipids inside of the membrane, disturb the lateral organization, and – in high concentrations – form mixed lipid/BAK micelles tearing out parts of the cytoplasmic membrane[61–64].

We performed HS-AFM imaging of the *S. aureus* lipid membranes exposed to BAK (Fig. 6, Supplementary Movie 6) and ethanol (Fig. S13). For BAK (Fig. 6, Supplementary Movie 6), we observe activity strikingly similar to AMC-109. Domains are clustered, which is followed by their dissolution and expansion of the membranes. From the molecular point of view, both AMC-109 and BAK thus seem to similarly incorporate in between the individual lipids in the *S. aureus* lipid membranes and disturb its lateral organization clustering and dissolving the lateral domains. In the case of ethanol we observe slowing down of the lateral movement of the domains and their fast dissolution. The effects of ethanol are very fast (Fig. S13), which points to a low kinetic barrier for the incorporation of the small ethanol molecule into the lipid membrane. We hypothesize that the bigger BAK molecules have to orient themselves prior to insertion into the membrane. Similarly, AMC-109 have to dissolve their aggregates into the membrane before acting inside the membrane. For both AMC-109 and BAK, this represents a kinetic barrier making them acting slower than ethanol.

BAK has been used widely for more than 70 years as a disinfectant and an antimicrobial in agriculture and industrial production, including for instance antimicrobial soaps[67]. It's widespread use led to partial reduced susceptibility in bacteria but full resistance has not been observed. Disinfectants are indeed powerful molecules, however, they cannot be used as antibiotics in human medicine for their toxicity to eukaryotic cells[68–71]. AMC-109 with its high selectivity for bacteria is not observed to be cytotoxic[11]. Moreover, the similarities with the BAK and ethanol mode of actions suggest that for AMC-109 it might take a while before resistance is generated in bacteria. A lower probability of resistance generation is further supported by the reported activity of AMC-109 against a number of *S. aureus* clinical isolates, including MRSA (96 strains), VISA (33 strains), and VRSA (13 strains)[14]. AMC-109 is, hence, capable of transport to the *S. aureus* membrane even through a thicker cell wall of resistant strains.

## Unique combined mechanism of AMC-109

In conclusion, AMC-109 uniquely combines multiple mechanisms in the fight against bacteria. AMC-109 monomers first form stable aggregates, which selectively interact with the negatively charged surface of bacteria. AMC-109 in its monomeric form is highly hydrophobic so it tends to interact with all membranes, including those of mammalian cells. The aggregation, in which the hydrophobic part of the AMC-109 molecule is hidden inside the aggregate and only the positively charged residues are exposed to water, is thus a crucial step responsible for the drug's high selectivity for bacteria and the absence of toxicity for human cells. The aggregates then attach to the bacterial cytoplasmic membrane and gradually dissolve all the AMC-109 individual molecules into the outer membrane leaflet. Once inside the membrane, AMC-109 affects the membrane lateral organization and lipid order, dissolving membrane domains. Functional membrane microdomains (FMMs) are important in *S. aureus* cells for protein sorting, signalling, or cell wall synthesis, as well as for the resistance of MRSA for penicillin, methicillin and other β-lactam antibiotics. Their dissolution takes all these functionalities away, leads to overall stiffening of the membrane, and possibly makes the MRSA colony again susceptible to β-lactams. The fact that AMC-109 targets negatively charged lipids, a basic feature of bacterial membranes, together with its similarities with the activity of the disinfectant BAK makes this antimicrobial agent particularly promising in not generating resistance in bacteria for a longer time. The high selectivity of AMC-109 for bacteria over human cellular membranes[11], provided by its aggregating properties, at the same time supports safer usage in medical treatment. We performed study on membranes from *S. aureus* lipid extracts, however, the findings on the AMC-109 activity could be applicable to a wide range of Gram-positive bacteria, as well as cancer cells, which also possess a net negative surface charge. By deciphering the molecular mode of action of AMC-109 we identified lateral membrane organization as a viable drug target, which can be used as a target in further pharmaceutical development. Dissolving membrane domains makes AMC-109 a highly selective disinfectant that is active against *S. aureus* bacteria, including multiresistant strains and possibly makes them again susceptible to treatment by β-lactam antibiotics like penicillin or methicillin.

# Methods

## Materials

AMC-109 (previously LTX-109[11]; Fig. 1a) was obtained in the form of dry powder from Amicoat AS, Norway. The powder was dissolved in milliQ water to stock solutions of 1–25 mg/ml, stored at room temperature shielded from the access of air and light, and used for the maximum of one month.

*Staphylococcus aureus* cells, strain RN4220, NCTC8325-4 derivative, restriction deficient and cured of prophages[72], was a generous gift from prof. J. M van Dijl from University Medical Centrum Groningen. The cells were stored in −80 °C.

Phosphate buffered saline (PBS buffer, pH 7.4) in the form of tablets for dissolution in milliQ water was purchased at Sigma-Aldrich. Synthetic lipids 1-palmitoyl-2-oleoyl-sn-glycero-3-phospho-(1'-rac-glycerol) (POPG), 1-palmitoyl-2-oleoyl-glycero-3-phosphocholine (POPC), and 1-palmitoyl-2-oleoyl-sn-glycero-3-phosphoethanolamine (POPE), di-oleoyl-phosphatidylglycerol (DOPG), di-oleoyl-biphosphatidylglycerol cardiolipin (DODOCL) and/or mono- and di-galactosyldiacylglycerol (MGDG and DGDG) were purchased from Avanti Polar Lipids as 25 mg/ml solutions in chloroform.

Pore-forming peptide melittin was purchased from GenScript Biotech (Netherlands) B.V. in the form of dry powder. The powder was dissolved in DMSO to stock concentration of 25 mg/ml and stored in −20 °C. The stock solution was diluted in miliQ water to 1 mM solution and used for one day. Final DMSO concentration during the AFM and leakage experiments was <0.010 vol% and <0.015 vol%, respectively. Test experiments showed no effect of DMSO on the membranes at these concentrations (Fig. S14).

Benzalkonium chloride (BAK), a mixture of molecules with acyl chain length varying between 8 to 18 carbons, was purchased in the form of semisolid gel with purity ≥ 95% from Sigma-Aldrich. The semisolid gel was dissolved in miliQ water to the stock

concentration of 1 g/ml. Ethanol with purity ≥ 97% was purchased from Boom B.V.

For the lipidomic studies: ammonium formate, 1-butanol, alpha-naphtol, molybdenum blue and ninhydrin were purchased from Sigma-Aldrich. Acetonitrile was purchased from Biosolve (Netherlands) B.V. For cobalt-calcein leakage assays: calcein, ethypenediaminetetraacetic acid (EDTA), $H_2KPO_4$, $HK_2PO_4$, and $CoCl_2$ hexahydrate with purity >98%, and the filtration gel Sephadex G75 were purchased from Sigma-Aldrich.

### S. aureus cells cultivation

S. aureus cells were cultivated in a growth medium in 37 °C, while shaking at 200–250 rpm, until the saturated growth phase. The cells were centrifuged at 2095 x g (centrifuge Allegra X-15R by Beckman, swing-out rotor SX4750A) at 4 °C for 10 min and the growth medium was exchanged for PBS buffer. The centrifugation step was repeated again and the excess PBS buffer was taken out. Wet cells were stored at −80 °C until lipid isolation.

### Lipid isolation from S. aureus wet cells

The lipids were extracted following the modified Bligh & Dyer extraction protocol[18]. In short, the cells were first washed in milliQ water and the pellets were transferred into tubes and weighted. The cells were dissolved in water/chloroform/methanol in ratios 0.8:1:2, where the water content was 0.8 ml per 1 g of the cells, and left stirring in 4 °C overnight. The mixtures were then centrifuged at 900 x g at 4 °C for 15 min. The supernatant was resuspended in a 1:1 chloroform/water mixture and left to phase separate at room temperature for 2 days. The bottom layer was collected and dried in a rotary evaporator. Finally, the dry lipid film was weighed and dissolved in chloroform to the stock concentration of 10 mg/ml and stored at −20 °C.

### Lipidomic analysis of the S. aureus lipid extracts

Lipid extracts were analyzed by ultra-high performance liquid chromatography-mass spectroscopy (UHPLC-MS) for the determination of the presence of various lipid species and their saturation state as described in more detail elsewhere[73,74]. In brief, 15 μl of a lipid extract (10 mg/ml in methanol) was injected on a Waters Acquity UPLC CSH C18 column (2.1 ×150 mm, 1.7 μm; Waters, cat#: 186005298) held at 55 °C. Lipids were separated using a constant flow of 300 μl/min and an elution gradient starting at 95% mobile phase A (5 mM ammonium formate in water:acetonitrile, 40:60) and 5% mobile phase B (5 mM ammonium formate in water:acetonitrile:1-butanol, 0.5:10:90) which was kept constant for the first 2.5 min. The proportion of mobile phase B was then linearly increased to a maximum value of 90% over 36.5 min and held for 3 min. This was followed by returning to 5% mobile phase B over 0.5 min which was then held for 8 min.

The UHPLC-MS instrument used consisted of an Accela1250 UHPLC system (Thermo Fisher Scientific) coupled to a Thermo Exactive Orbitrap mass spectrometer (Thermo Fisher Scientific) equipped with an ESI ion-source in negative ionization mode. The spray voltage was set at 3.0 kV, whereas the capillary, tube lens and skimmer voltage were set to −75 V, −190 V and −46 V, respectively. The capillary temperature was set at 300 °C. Sheath- and aux-gas flow rate was set at 60 and 5 units, respectively. To obtain fragmentation MS spectra, the CID function with an energy setting of 80 eV was used. A scan range of $m/z$ 250–2000 was used for full-MS spectrum acquisition and a range of $m/z$ 77.5–1550 for fragmentation spectrum acquisition. Data was analyzed and extracted using the Thermo Xcalibur Qual Browser (Thermo Fisher Scientific).

Relative quantification of glycerophospholipids and glycolipids was performed by spotting 1, 5 and 10 μl of the lipid extract 2 cm from the bottom on aluminum-backed silica gel 60 TLC plates (cat#: 1.05553.0001; Merck) together with DOPG, DODOCL and/or MGDG and DGDG reference compounds. Lipids were separated using a mobile phase consisting of chloroform:methanol:glacial acetic acid (65:25:8) until the solvent front reached a point approximately 1 cm from the top of the TLC plate. The TLC plates were first dried at RT and then stained by momentarily horizontally dipping them in a shallow pool of staining solution: (a) molybdenum blue staining solution for the visualization of phosphate-containing lipids, (b) 0.5% (w/v) α-naphtol in ethanol or 1-butanol with $H_2SO_4$ (9:1, v/v) (adapted from[75] to visualize glycolipids or (c) 0.1% (w/v) ninhydrin in water-saturated 1-butanol[76] to visualize lipids containing primary or secondary amines. The TLC plates were then left at room temperature to develop and heated with a hairdryer if necessary.

For relative quantification, the TLC plates were digitized using a PowerLook 1120 scanner (Amersham Biosciences) in combination with the VueScan software (v9.094, Hamrick Software). Images were analyzed using the Fiji[77] "gel analyzer" tool (build: ImageJ V1.53c) to produce chromatograms and calculate integrated peak values from which relative values were calculated.

### Atomic force microscopy experiments

Preparation of liposomes: S. aureus lipid extracts in chloroform were pipetted into the glass vial. The chloroform was evaporated using an argon stream and let dry completely for >1 h in a vacuum. PBS buffer was added to the lipid film in the concentration of 0.3 mg/ml. To help the resuspension, the mixture was vigorously shaken for 1 minute followed by five cycles of freezing by liquid nitrogen and thawing in warm water. The resuspended liposomes were then extruded 21 times through 0.1 μm pores polycarbonate membranes (Avanti Polar Lipids). The liposomes were stored in 4 °C and used for the maximum of one week.

Sample cell for AFM was prepared by gluing a small piece of mica on microscope glass slide with a transparent epoxy glue. A glass ring was attached around it by two-component biocompatible glue (Bruker Nano GmbH). The liposomes were diluted to 0.02–0.04 mg/ml in PBS buffer. 10 μl drop of the diluted liposomes was deposited on freshly cleaved mica surface and let to sediment for >10 min. In some cases, the PBS buffer was washed out 5 times and replaced with a new 10 μl drop of PBS buffer to press the liposomes on the mica surface and support the rupture of the liposomes and formation of isolated supported membrane patches. PBS buffer was then added to the total volume of 1 ml.

Supported membranes made of S. aureus lipid extracts are denoted as S. aureus lipid membranes in the text.

AFM imaging and nano-indentation of the supported S. aureus lipid membranes immersed in PBS buffer was performed with a JPK Nano Wizard Ultra Speed AFM. The experiments were performed at room temperature (22 °C) using qp-BioAC cantilevers (NanoAndMore GmbH) with a nominal spring constant 0.06 ± 0.03 N/m and a silicon nitride tip with a typical tip radius of curvature smaller than 10 nm. The imaging force was ~80–100 pN in all cases.

During the nano-indentation experiment (Supplementary Note 1, Fig. S5), the AFM tip approaches the supported membrane in a constant velocity and indents the membrane until reaching the underlaying support (Fig. S5a). First, the membrane was imaged, then the hard mica surface was indented with a force up to 2 nN with an indentation velocity of 300 nms⁻¹ to check for any tip contamination and get a hard surface reference, which gives us the bending behavior of the cantilever itself. Immediately after the check of the tip contamination the membrane was indented with a force up to 1 nN with an indentation velocity of 300 nms⁻¹. The dependence between the force applied on the membrane $F$ and the vertical tip position $z$—the force-indentation curve—is recorded. If multiple spots on membrane were indented, clean surface curve was measured before each new membrane indentation. After indentations, the same spot was imaged again. For AMC-109 concentrations >2 μg/ml, in which the membrane

expands over the substrate, nano-indentation experiments were not possible due to adhesion forces in between the AFM tip and the surface. This was probably due to AMC-109 molecules covering both the expanded membranes and the surrounding mica surface. We hence focused our investigation on the mechanical properties of *S. aureus* lipid membranes exposed to AMC-109 concentrations <2 μg/ml. At 1 μg/ml in part of the membranes the lateral domains are completely dissolved, whereas in some membrane patches they are still visible. There is also a part of the membranes where we could not make a clear distinction between these two cases. Moreover, it is not possible to reliably distinguish from the force-indentations curves if we are indenting a region on top of the domain or out of it. Mechanical properties shown in Fig. S5 were, hence, calculated from all data, gathered both on top and out of the lateral domains.

Both the AFM images and the force curves were processed using JPK Data Processing Software. AFM force-indentation curves analysis: In brief, the nano-indentation experiment yield the deflection of the cantilever as a function of the Z-piezo displacement. This dependency was converted to the applied force versus the vertical distance between the tip and the sample surface by subtracting the deflection of the cantilever itself[78], which was determined from the force-distance curves recorded on a hard surface. The penetration point, when the tip perforated the membrane was determined as a point where we observe an apparent drop in the applied force. The region between the first tip–membrane interaction and the full membrane penetration was fitted with a modified Hertz model using Equation S1 (Supplementary Note 1) according to a modified Hertz model of a thin supported layer with a 1 nm water layer bellow[58]. Two parameters were derived from the fit: vertical distance of the first tip–membrane interaction related to the membrane height and Young's modulus of the membrane describing its tensile stiffness. The analysis of the mechanical properties of the *S. aureus* lipid membranes in the presence of 0, 0.5 and 1 μg/ml AMC-109 was done independently on 150, 119, and 144 individual force-indentation curves, respectively. The resulting Young's modulus values (Fig. S5) are reported as median ± standard error of the mean.

## High-speed atomic force microscopy

Liposomes made from *S. aureus* lipids were prepared for the HS-AFM experiments the same way as for AFM. Synthetic liposomes were prepared in concentrations 1 mg/ml in PBS. In cases where the attachment of membranes to mica was weaker, we exchanged the last extrusion step in the liposomes preparation with gentle sonication for 30 s in a sonication bath prior to the liposomes deposition on mica. Expansion of synthetic lipid membranes was measured at POPC, DOTAP/POPC (60/40 mol%), and POPG/POPC (60/40 mol%).

HS-AFM experiments were done using RIBM (Japan) machine in amplitude modulation tapping mode in liquid[26,27,79,80]. Short cantilevers USC-F1.2-k0.15 (NanoWorld, Switzerland) with a spring constant of 0.15 N/m, resonance frequency around 0.6 MHz, and a quality factor of ~2, and ultra-small cantilevers BL-AC10FS-A2 (Olympus, Japan) with a spring constant of 0.1 N/m, and resonance frequency around 0.6 MHz in buffer were used. The cantilever-free amplitude was set to 1 nm, and the set-point amplitude for the cantilever oscillation was set around 0.8 nm. Images were taken at 300 ms to 6 s per frame depending on the size of the image. A mica surface of diameter 1.5 mm glued on top of a 5 mm high glass rod was used as the AFM sample stage. The glass rod was then attached to the scanner Z-piezo using a small amount of nail polish. For the activity of the antibiotic agents (AMC-109, melittin, ethanol, BAK) on synthetic and *S. aureus* lipid membranes, a 3 μl drop of 0.03–0.5 μg/ml liposomes in PBS buffer was deposited on the freshly cleaved mica and the buffer was washed out 5 times to press the liposomes on the mica surface and support the rupture of the liposomes and formation of isolated supported membrane patches (as in AFM experiments). The scanner head was then put

upside down into a small liquid chamber containing the cantilever and filled with 40–80 μl of the recording solution (PBS buffer). First, we image untreated membranes and then add 5 μl of the buffer with a concentrated solution of the active agent to achieve the desired final concentration in the sample.

Size distribution of lateral lipid domains was measured at 269 individual domains across multiple experimental days. They showed diameter of 31 ± 6 nm. Membrane thickness was measured by comparison of the measured height at the parts outside of the lateral domains with the height measured at the adjacent mica surface. Similarly, the height of the AMC-109 aggregates was measured as a difference between the height measured on top of the aggregate and on the surrounding mica. Expansion of the synthetic lipid membranes after the addition of 4 μg/ml AMC-109 was measured from 3–4 individual experiments per lipid composition, in which expansion of 7–14 membrane patches was analyzed. Height, thickness, membranes expansion, and domains size measurements from AFM and HS-AFM images are reported as mean ± standard error of the mean.

## Fluorescence leakage assays

In cobalt-calcein fluorescence leakage experiments, calcein is trapped inside of the liposomes, where its fluorescence is quenched by the formation of complexes with cobalt ions (Fig. 5a). The addition of a perforating agent to the outside buffer leads to the leakage of cobalt-calcein complexes out of the liposomes, where cobalt is extracted from the complex by ethylenediaminetetraacetic acid (EDTA) resulting in an increased calcein fluorescence intensity.

50 mM and 35 mM phosphate buffers, pH 7.0, were prepared as mixtures of $H_2KPO_4$ and $HK_2PO_4$. 35 mM phosphate buffer was mixed with 10 mM EDTA and pH was adjusted to pH 7.0. POPG/POPC/POPE lipids in ratios 60:30:10 mol% were pipetted from the stock solutions in chloroform into the glass vial. Chloroform was evaporated using argon stream and let to completely dry out for > 1 h in vacuum. The lipid film was rehydrated using a mixture of 1 mM $CoCl_2$ and 0.8 mM calcein dissolved in 50 mM phosphate buffer, pH 7.0, into the lipid concentration of 100 mg/ml. Resuspension was promoted by five cycles of freezing in liquid nitrogen and thawing in warm water. Formed liposomes were extruded 21 times through 0.4 μm pores polycarbonate membranes (Avanti Polar Lipids). The extruded liposomes were stored in 4 °C overnight.

On the day of the experiment, the extruded liposomes were washed from the external buffer containing free $CoCl_2$ and calcein. The original buffer was replaced with the 35 mM phosphate buffer with 10 mM EDTA, pH 7.0 using gravity separation column with the internal volume of 12 ml packed with Sephadex G-75. The column was first preequilibrated with the 35 mM phosphate buffer with 10 ml EDTA, pH 7.0, and then used for separation of the extruded liposomes from the original external buffer and the free calcein and $CoCl_2$. All fractions containing slightly fluorescent liposomes were collected into a single tube and ultracentrifuged at 444 000 x g for 25 min at 4 °C. The buffer on top of the pellet of liposomes was removed, and the pellet was resuspended in fresh 35 mM phosphate buffer with 10 mM EDTA, pH 7.0 to the stock concentration of 100 mg/ml.

The fluorescence measurements were performed with Fluorometer QuantaMaster 40 (PhotoMed GmbH). The liposomes were diluted to 0.1 mg/ml for the experiments. The calcein was excited at $\lambda_{ex}$ = 495 nm and the emission at $\lambda_{em}$ = 520 nm was detected in time with both the excitation and emission slits opened to the range of wavelengths ± 5 nm. Background fluorescence was measured for 5 min, then membrane-active agent (AMC-109 or melittin) was added and the increase in the fluorescence intensity from the calcein leaking out of the liposomes was detected for 10 min. Finally, detergent Triton X-100 in the final concentration of 0.25 vol% was added to rupture all the liposomes and release all of the calcein. The maximum intensity of the fluorescence signal was detected for 10 min. The time traces of the

fluorescence intensity were normalized to 0 at the background intensity before the addition of the membrane-active agent and to 1 at the maximum intensity after the addition of Triton X-100.

## Molecular dynamic simulations

The initial configurations of the simulated lipid bilayers were generated using the tool Insane[81] to yield lateral dimensions of 30×30 nm from 3042 lipids in total with the bilayer repeat distance of 30 nm. To distinguish between eukaryotic (neutral) and bacterial (negatively charged) membranes, we used a simple mixture of POPC/POPG lipids, with the anionic POPG fraction varied between 0–100 mol%. Using the GROMACS tool *gmx solvate*[82], 2489 AMC-109 molecules were distributed in a water solution with a minimum distance of 1 nm between each of them. The resulting molar ratio of AMC-109:water molecules was 1:250 (approximately 4 water molecules are represented by 1 Martini bead). The total charge of the system is neutralized by Chloride counterions with an additional 100 mM NaCl concentration in the solution. A typical simulation setup is illustrated in Fig. S8.

The simulation temperature was coupled to a v-rescale thermostat[83] at room temperature of 310 K, separately for lipids, AMC-109 and solvent. A Parinello-Rahman barostat[84] was used for pressure coupling at 1 bar with a coupling constant of 24 ps independently for the membrane plane and its normal. The standard time step of 20 fs was used for all simulations and the trajectory was recorded every 1 ns. All systems were equilibrated for at least 100 ns prior to production simulations. For production, we have simulated all systems for at least 7 μs to ensure convergence. The simulation settings used in this work agree with the standard settings for MARTINI 3 model[38] implementation in GROMACS.

The topology for AMC-109 was first generated based on the corresponding peptide template, using the protocol outlined in ref. 38 and the resulting peptide topology was subsequently modified to include the modification in the Tryptophan residue and the changes to the C-terminus as outlined in Ref. 11. The topologies for lipids, ions and solvent were taken directly from the Martini 3 collection[38].

Simulations were run using GROMACS simulation package ver. 2019.3 in a mixed precision compilation without GPU support[82]. The profiles of the number of contacts between AMC-109 and lipids were calculated using GROMACS tool *gmx mindist* with a cutoff radius of 0.6 nm between the groups of molecules. Scripts used to analyze the simulations, the molecular topology of AMC-109 and simulation setup files in GROMACS format, and trajectories of the lipid bilayers with AMC-109 molecules are available in a public repository[85].

## Reporting summary

Further information on research design is available in the Nature Portfolio Reporting Summary linked to this article.

# Data availability

The data that supports the findings of this study can be found in the manuscript, its Supplementary Information, and provided Source Data file. Unprocessed AFM images, force-distance curves used for mechanical characterization, and raw mass spectroscopy data are available in an open public repository[86]. Scripts used to analyze the simulations, the molecular topology of AMC-109 and simulation setup files in GROMACS format, and trajectories of the lipid bilayers with AMC-109 molecules are available in a public repository[85]. Unprocessed HS-AFM data and data from experimental repetitions are available from the corresponding author upon request. Source data are provided with this paper.

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

## Acknowledgements
WHR thanks the EU for support through the INFRAIA infrastructure grant MOSBRI. AM thanks the Nederlandse organisatie voor Wetenschappelijk Onderzoek (NWO) for support through a Physics/f grant (no. 680-91-007) and an XS grant (no. OCENW.XS22.1.083).

## Author contributions
A.M., S.M., J.M. and W.H.R. have made a substantial contribution to the concept and design of the work; A.M., S.M., J.M., N.A.W.dK., M.G., and Jv/dE. took part in the data collection, data analysis and interpretation; A.M. drafted the article; A.M., S.M., J.M., N.A.W.dK., W.S., J.S.M.S., A.J.M.D., S.J.M., and W.H.R. have critically revised the article; All the authors have approved the final version to be published.

## Competing interests
JSMS and WS are employed by Amicoat AS, the producer of AMC-109. The remaining authors declare no competing interests.
