## [Peer Review File · Nature Communications]

Lateral membrane organization as target of an antimicrobial peptidomimetic compoundReviewer #1 (Remarks to the Author):

The manuscript reports studies of an amphiphilic compound (i.e. AMC-109) modulating bilayer structures of synthetic lipids and lipids extracted from bacteria. The techniques employed include AFM, high-speed AFM, and coarse-grained simulations, as well as leakage experiment. The results are intriguing and important. There are several concerns as discussed below. The authors may also want to check references so more relevant works are cited (e.g. ref. 45 is not a good resource for melittin induced leakage).

Bilayer area expansion after the addition of AMC-109 is shown in Fig. 1. As the concentration increased to 3 $\mu\text{g/mL}$, the area coverage is drastically increased from 9 to 71%. The authors need to address what is contributing to such a large area increase (e.g. what is the possible molecular arrangement of lipids and AMC-109 in the expanded bilayer structure?). Assuming the number of lipids does not change, the increased area is mainly contributed by AMC-109 in a form similar to a bilayer organization. Although the authors did show that AMC-109 aggregates can be adsorbed to mica, the thickness of AMC-109 aggregates varies significantly (1-3 nm) compared to the figure shown in Fig. 1.

Regarding the bilayer area expansion, isolated patches were prepared on mica. The presence of bilayer boundary allowed the expansion of bilayer organization laterally. In bacteria, such lateral expansion is inhibited because the membrane is continuous. The authors may want to address how the area expansion observation is relevant to the antimicrobial mechanism.

Melittin forms pores with diameters less than 10 nm. Such AFM data have been reported earlier. Proper reference should be cited. The authors used DMSO to prepare melittin stock solution. This is quite unusual as melittin is water soluble. For the kinetic leakage study of AMC-109 and melittin, the incubation time of AMC-109 with liposome should be extended much longer compared to melittin. This is because AMC-109 tends to aggregate which melittin does not.

The AFM force of 80-100 pN seems very small compared to typical values of ~ 300 pN reported by other groups. The small force could be related to the small puncture force of ~ 200 pN shown in Fig. S5. Any explanation to why the bilayer puncture force is so small compared to reported values by various groups (>1 nN)?

Coarse-grained simulation was performed. Dynamic events of aggregate insertion were shown. However, no analysis was performed regarding the impact on lipid structure and order. This compromises the goal of this report, which is to determine the mode of membrane perturbation by AMC-109.

It seems that HS-AFM is capable of imaging all bilayer structures in this report. Is there any reason to use traditional AFM in this report (e.g. Fig. 1)?

Reviewer #2 (Remarks to the Author):

Melcrova and colleagues show in this work the potential mechanism of action of the antibacterial peptidomimetic compound AMC-109. This peptide is already being tested for antibacterial activity against skin infections and nasal colonization but surprisingly, the mechanism of action has not been elucidated yet. In this work, they suggest that AMC-109 causes an alteration of the lateral membrane organization that reduces bacterial viability. The authors claimed that this is specific, or more selective, to bacterial membranes than eukaryotic membranes. Overall, this is a very promising line of research and the potential antibacterial activity of AMC-109 may develop into a new class of antibiotic. It may also open a new venue to selectively kill antibiotic-resistant bacteria by altering bacterial membrane organization thus affecting many cellular processes at once.

There are three important aspects of the mechanism of action of AMC-109 that are not sufficiently addressed in this work and need to be developed further to clarify how AMC-109 works. One is

related to the selectivity of AMC-109 against bacterial membranes against eukaryotic membranes; the second aspect is related to the activity against entire bacteria rather than purified membranes or artificial membranes. The third aspect is related to how fast would the resistance against AMC-109 occur in bacteria that are exposed to the antibiotic. This third aspect is key in the microbiology field to elucidate/confirm a potential mechanism of action of any given antibiotic.

1) How selective is AMC-109 against bacterial membranes? The authors claimed that AMC-109 is more selective against negatively-charged membranes of bacteria than neutral eukaryotic membranes. This is an interesting point but it needs to be validated *in vitro* and *in vivo*. For instance, bacteria have a cell wall that acts as a barrier against many molecules. It may require a concentration higher than expected to penetrate the cell wall and probably at this concentration is more toxic to exposed eukaryotic membranes. In addition, even if the compound does not affect the membrane organization of eukaryotic membranes, it would be important to know whether it integrates in eukaryotic membranes and forms aggregates. This is important because the accumulation of aggregates in cellular is known to cause severe pathologies.

2) The activity against entire bacteria rather than purified membranes or artificial membranes. Most of the work involved in the characterization of AMC-109 has been performed using purified membranes or artificial membranes. To be clear, this referee does not criticize the experiments and modelling presented in this work but it should be combined with experiments using living bacteria. Again as an example, bacteria have many ways to maintain membrane homeostasis and probably they have potential mechanisms to counteract the effect of AMC-109 that are not known yet. It is important to validate the *in vitro* results using purified membranes in living cells.

3) Evolution of resistance against AMC-109. The authors claimed that it might take a while before resistance is generated in bacteria. This is possible and would be fantastic but this needs to be tested. It is actually an important step for a microbiologist to elucidate the resistant mechanism (or at least to identify mutations that confer resistance) because from these mutations one can validate the mechanism of action of the compound. Possibly the resistance mechanism is based on preventing the contact of AMC-109 with the bacterial membrane by increasing the thickness of the cell wall, similar to the intermediate-resistance mechanism to vancomycin of *S. aureus* clinical strains (VISA). The authors at least should try how effective is AMC-109 against a VISA strain (a model VISA strain such as Mu50) because this may support the notion that AMC-109 needs to pass the cell wall to be active and thus is consistent with the hypothesis that it attacks lateral membrane organization.

Minor comments

Why the authors focus on membrane damage on the first time? There is no initial indication that AMC-109 caused membrane damage. How they concur that the membrane is the target?

The authors claimed that a single membrane leaflet and its effects on the lateral organization in fact resemble the activity of small disinfectant molecules like ethanol, benzalkonium chloride (BAK) or benzyl alcohol. These are used as disinfectants but they cause damage in eukaryotic cells. A comparative study of damage in bacterial and eukaryotic cells is needed in this work.

The authors claimed that AMC-109 is not a pore former using a pore-former molecule melittin as control. Totally honest, I have never heard of this pore former. What are the differences between this pore former and other classical pore formers such as nisin or colistin?

Reviewer #3 (Remarks to the Author):

The authors provide data that sheds light on the functional mechanism of the known peptide AMC-109. Although the experimental work provided is relatively sound, the article lacks the level of

novelty expected for Nature Communications.

Specific comments:

- Peptide AMC-109 has been described previously, significantly decreasing the novelty of this study.
- AMC-109 should be referred to as a peptide, not a peptidomimetic, since the backbone is composed of amide bonds and the alpha carbons as in a regular peptide. Non-canonical residues do not necessarily make it a peptidomimetic, and the capped N-terminal does not warrant this classification either in my view. Examples of peptidomimetics would be peptoids, peptides composed of beta-amino acids, etc. The arginine residues are not arginine-like residues, they are arginine residues.
- All concentrations should be indicated in μM , which make AMC-109 very average in terms of bioactivity.
- The authors should discuss whether a less hindered Trp derivative would be beneficial or not for the proposed mechanism of action. The tert-butyl groups would only make it difficult for the peptides to aggregate and would delay the kinetics, which seems to be important for mechanism of action. The Trp analog would make a good control.
- According to the aggregate size distribution results, please discuss if this mechanism of action is expected for an aggregate which is almost the size of a lipid bilayer (3-4 nm in length).
- Why are the MD simulations done in membranes that are different from those found in the lipidomics study? POPC/POPG (MD) vs CL/PG/Lysyl-PG (lipidomics). The same comment applies for the calcein leakage studies (POPG/POPC/POPE).
- How translatable are the results obtained in a 2D lipid coating experiment compared to the membrane of a microorganism, considering tension and other physical parameters that will most likely influence how the lipid bilayer will respond to the peptide or aggregate upon contact?
- Why was melittin used as a comparative control when it has a completely different mechanism of action (pore-forming)?
- Please add positive controls for the first dissolution of lateral domains experiments.
- More recent references should be added.

Reviewer #4 (Remarks to the Author):

This is a nice, mainly AFM study on the antimicrobial peptidomimetic tripeptide AMC-109, developed more than a decade ago and extensively tested in clinical trials since. The authors do lipidomics to come up with a synthetic model of the membrane of *S.aureus* (w/o peptidoglycan) and use that in the AFM experiments.

The AFM pictures look beautiful and it is convincingly demonstrated experimentally that AMC-109 adsorbs into these membranes, expands them laterally, dissolving lipid rafts and thinning the membrane overall.

That this is the way *S.aureus* is killed is conjectured from two controls: The same AFM methods are used on the toxic honey bee peptide melittin, which – in contrast - shows poration as the mechanism. This is also corroborated by a calcein leakage assay that shows poration for melittin and not for AMC-109. The second comparison is to the small disinfectant molecule benzalkonium

chloride (BAK), which shows results very similar to AMC-109. So the mechanism is claimed to be a change in membrane lateral organization and lipid order, and dissolving membrane domains. It is hypothesized that the bacterium cannot survive if its functional membrane microdomains (FMMs) are dissolved, although no direct evidence is gathered for this here as the experiments are purely on synthetic bilayer constructs, not organisms.

A bit curious is the addition of coarse grain (CG) MD simulations as a way to visualize what is going on. Firstly, I'd like to point out that matching the results of experimental assays and MD simulations is a very challenging task, so the authors should be commended for attempting it. However, here I have some observations that appear incongruous:

- The whole manuscript is focused on lipid domains and their dissolution, as well as the overall membrane thinning due to incorporation of AMC-109. Yet, the CG MD simulations do not show any of this. No lipid domains are visible. And no membrane thinning is demonstrated.
- The CG MD uses POPC/POPG bilayers. Significant lipidomics was done to arrive at a simple model of the membrane of *S.aureus*. Why did the CG simulations not simply employ those lipid compositions?
- CG MD predicts that AMC-109 aggregates are adsorbed and dissolved after 300 ns - 2 μ s (Figure 4a). After that, individual AMC-109 molecules are laterally diffusing among the lipids in the upper bilayer leaflet and a steady state appears reached. Yet the experiments show timescales of 4 min for domain dissolution and 15 min for bilayer thinning. That is an 8 order of magnitude difference. Could the authors give an explanation?
- A nice result is the demonstration that AMC-109 aggregates in solution, and that the AFM measurements seem to see and corroborate the size of these aggregates (Figure 3).
- When these aggregates are then let loose to adsorb on the membrane, they are seen selective in binding for PG and not PC. CG force fields have very smooth energy landscapes that are ideally suited to study aggregation, adsorption or other phenomena with large, homogeneous phases, such as lipid membranes. The effect seen here (and shown many times before with other aggregates on membranes) is the rapid minimizing of the free energy due to charge compensation (i.e. positively charged CG beads get close to negatively charged CG beads). As the authors show, there is a direct correlation of AMC-109 adsorption with the concentration of anionic PG lipids, so that one AMC-109 (charge +3) adsorbs per 3 PG (charge -1) until the membrane is neutral and then stops. Why? What happens to the remaining AMC-109 molecules? Do they stay in solution? In this case, a neutral bilayer would not be the preferred free energy minimum location of AMC-109, but rather it prefers the a hydrophobic self-aggregate instead of the hydrophobic membrane core as a way to bury its hydrophobic parts. So it stays in solution as hydrophobic aggregate when membranes are neutral and only loves anionic membranes.
- This seemingly explains the lack of insertion in the PC bilayer. However, the reason is then not a kinetic one, as the authors speculate (quote: "Second, self assembly of AMC-109 creates an energetic barrier preventing interaction of AMC-109 with neutral membranes"). Given the smoothness of CG force fields, which lack hindering hydrogen bonds slowing kinetics, this does not sound too convincing. It could be checked by just putting AMC-109 molecules into the PC bilayer from the start and seeing if it they stay there. This would make the bilayer positively charged, but the counterions compensate. The current results are puzzling, because in the HF-AFM measurements, AMC-109 clearly binds to pure POPC and increases its surface by 16% (Figure 4C), whereas in the CG MD, it does not bind at all. Even more puzzling: If the membrane is made positively charged (POPC/DOTAP), there is a significant 50% increase in surface area, indicating substantial AMC-109 adsorption. This seems to contradict the CG MD observations that insertion is merely proportional to anionic charge content due to 3:1 charge compensation. Clearly, AMC-109 also loves cationic membranes, and neutral ones a bit. The authors even mention this (quote: "AMC-109 in its monomeric form is highly hydrophobic so it tends to interact with all membranes including those of mammalian cells").

These differences in the HS-AFM results and CG MD are difficult to explain. I suspect that the mechanism derived from CG MD may be a bit too simplistic. If it is just a kinetic barrier, or just charge compensation, then any cationic peptide or petidomimetic, many of which also form aggregates in solution, would be a good antimicrobial or anti-cancer drug, as those targets have anionic membranes. However, most cationic peptides are cytotoxic to eucaryotic cell membranes, which are not negatively charged.

It would be nice if the authors made some more control simulations to address these discrepancies between HS-AFM and CG MD, and try to corroborate their theory that the lack of toxicity of AMC-109 is due to a kinetic barrier in unpacking the aggregates, not thermodynamic reasons.

Minor points:

- Fig. 3b: The picture of AMC-109 in CG representation looks really strange. Maybe a stick-model is not suited. It would be better to show how this molecule really looks in CG representation, i.e. CG beads or ball-and-stick model? So it can be judged how much information is missing as compared to an all-atomistic model.

- typo: l264 'Only a these very high concentrations' -> at

Rebuttal

Point-by-point response to reviewers' comments

Reviewer #1 (Remarks to the Author):

The manuscript reports studies of an amphiphilic compound (i.e. AMC-109) modulating bilayer structures of synthetic lipids and lipids extracted from bacteria. The techniques employed include AFM, high-speed AFM, and coarse-grained simulations, as well as leakage experiment. The results are intriguing and important. There are several concerns as discussed below. The authors may also want to check references so more relevant works are cited (e.g ref. 45 is not a good resource for melittin induced leakage).

We thank the reviewer for pointing this out and we checked all references.

We revised the following references:

- Ref. 45 was indeed placed there by mistake. We deleted it as it is not relevant for the manuscript. A new reference (Khadka et al. *BBA Biomembranes*, **1859**(5) (2017), <http://dx.doi.org/10.1016/j.bbamem.2017.01.026>) was added.
- Ref. 70 and Ref. 73 were duplicate. (Abraham et al. *SoftwareX* **1–2**, 19–25 (2015)). This was corrected and the reference is now used only once.

Bilayer area expansion after the addition of AMC-109 is shown in Fig. 1. As the concentration increased to 3 $\mu\text{g}/\text{mL}$, the area coverage is drastically increased from 9 to 71%. The authors need to address what is contributing to such a large area increase (e.g. what is the possible molecular arrangement of lipids and AMC-109 in the expanded bilayer structure?). Assuming the number of lipids does not change, the increased area is mainly contributed by AMC-109 in a form similar to a bilayer organization.

We agree that a specific discussion was lacking and is useful. As we need the MD results in order to understand the spreading, we added the discussion on the molecular rearrangement in the results section "Molecular picture of AMC-109 attacking model membranes". Fig. 4b shows the detail of the AMC-109 molecules inserting the phospholipid bilayer. After first attachment of the AMC-109 aggregate on the bilayer surface, individual AMC-109 molecules are inserted into the bilayer. They move laterally in between the phospholipids and do not form any further aggregates inside the bilayer. The number of inserted AMC-109 molecules is for negatively charged membranes very large (see Fig. 4a) resulting in a high increase in material that forms the membrane. Apart from this contribution, the membrane also gets thinner, which is reported by HS-AFM measurements in Fig. 2c. Thinning of the membrane together with increase in its material leads to significant spreading reported in Fig. 1. The manuscript is now updated with this explanation.

Although the authors did show that AMC-109 aggregates can be adsorbed to mica, the thickness of AMC-109 aggregates varies significantly (1-3 nm) compared to the figure shown in Fig. 1.

In Fig. 1 the whole area around the membrane is covered with a layer of AMC-109. This layer is formed by fusion of individual aggregates. In Fig. 1, both the background (mica, mica + AMC-109) and the membrane change their height. The background gets higher with the layer of AMC-109, and at the same time the membrane gets thinner as a consequence of AMC-109 insertion (Fig. 2c). Both these processes contribute to lowering the relative difference between the height of the membrane and its surrounding. As we cannot separate these two contributions, we opted to keep the color saturation on top of the membrane constant throughout the panels containing 1-4 $\mu\text{g}/\text{ml}$ AMC-109,

and change the hue only of the background. The changes in background color thus include both the AMC-109 height and the membrane thinning. We realize this might be misleading. An explaining sentence has been added to the caption of Fig. 1.

Regarding the bilayer area expansion, isolated patches were prepared on mica. The presence of bilayer boundary allowed the expansion of bilayer organization laterally. In bacteria, such lateral expansion is inhibited because the membrane is continuous. The authors may want to address how the area expansion observation is relevant to the antimicrobial mechanism.

We expect that in bacteria cells stress builds up in the membrane, which might eventually lead to partial membrane remodeling via the formation of rough surfaces and protrusions. We added a comment on this aspect to the manuscript.

Melittin forms pores with diameters less than 10 nm. Such AFM data have been reported earlier. Proper reference should be cited.

A sentence stating that our observations are in line with previous AFM studies has been added together with 2 relevant references. (Juhaniwicz, J. & Sek, S. *Electrochim Acta* 197, 336–343 (2016), <http://dx.doi.org/10.1016/j.electacta.2015.11.134>; Giménez, D., Sánchez-Muñoz, O. L. & Salgado, J. *Langmuir* 31, 3146–3158 (2015), <https://pubs.acs.org/doi/10.1021/la504293q>)

The authors used DMSO to prepare melittin stock solution. This is quite unusual as melittin is water soluble.

Melittin molecule is stable in water only for 24 hours. For our experiments we use very small amounts of melittin (max 5–25 μ l of 40 μ M). Dissolving this small amount in water directly from powder is technically challenging/not possible. We prepared a 8.78 mM stock in DMSO for long term storage in -20 °C. Every experimental day, we make a fresh dilution in extra pure water that we use for maximum of 24 h. The residual DMSO concentration in the sample is <0.02 vol%. Storing a stock of melittin in DMSO was also for instance reported by Giménez, D., Sánchez-Muñoz, O. L. & Salgado, J. *Langmuir* 31, 3146–3158 (2015), <https://pubs.acs.org/doi/10.1021/la504293q>. The information on the residual DMSO concentration can be found in the materials and methods section. A test experiment showed that 0.02 vol% DMSO does not make any damage on *S. aureus* lipid membranes. We added the results of this test experiment as an additional figure to the supporting information (Fig. S14).

For the kinetic leakage study of AMC-109 and melittin, the incubation time of AMC-109 with liposome should be extended much longer compared to melittin. This is because AMC-109 tends to aggregate which melittin does not.

AMC-109 aggregates are already present in the stock of AMC-109 prior to their addition to liposomes loaded with fluorescent dye, so we do not need to wait until they are formed. According to our HS-AFM data, 10 minutes is long enough for AMC-109 aggregates to attach and dissolve into the membranes. HS-AFM data on membrane spreading show significant changes within the first 10 minutes (Fig. 4c–d). The domains dissolution in *S. aureus* lipid membranes is also observed in less than 10 minutes (Fig. 2b). The fact that in those 10 minutes no leakage is observed for AMC-109 concentrations up to 4 AMC-109 per single lipid, i.e. 400 μ g/ml AMC-109 clearly shows no poration activity.

The AFM force of 80-100 pN seems very small compared to typical values of \sim 300 pN reported by other groups. The small force could be related to the small puncture force of \sim 200 pN shown in Fig.

S5. Any explanation to why the bilayer puncture force is so small compared to reported values by various groups (>1 nN)?

We aim for as small imaging force as possible not to disturb the sample. We use cantilevers with a nominal spring constant 0.06 N/m (0.03–0.09 N/m), which is lower than usually used by other groups. This in combination with the optimization of imaging parameters probably leads to the smaller imaging force. We do not think this is related to the penetration force.

The penetration force mainly depends on the tip radius (<10 nm in our case) and the membrane composition. Comprehensive comparison of the tip size and membrane composition on the penetration force was done in study Saavedra et al. *Langmuir* **36**, 5709–5716 (2020) (<https://pubs.acs.org/doi/pdf/10.1021/acs.langmuir.0c00247>). They report penetration forces for a 2 nm tip to be ~100 pN at DOPC membrane, and ~1.5 nN for at DPPC membrane. For a 10 nm tip they report ~2 nN at DOPC membrane, and ~10 nN for at DPPC membrane. The variation is, hence, quite big.

Our results suggest that the AFM tips we use have indeed a diameter lower than 10 nm and we are indenting a soft, fluid membrane. Majewska et al. *BBA Biomembranes* **1862**(9) (2020) (<https://doi.org/10.1016/j.bbamem.2020.183347>) used the same cantilevers as used in our study. Their measurement of the tip sizes using SEM led to an average value of 8 nm that they used for mechanical analysis. We now added this information to the SI.

Coarse-grained simulation was performed. Dynamic events of aggregate insertion were shown. However, no analysis was performed regarding the impact on lipid structure and order. This compromises the goal of this report, which is to determine the mode of membrane perturbation by AMC-109.

MD simulations were performed to provide a molecular insight into the behaviour of AMC-109. Simulations revealed the micellar structure of the AMC-109 aggregates allowing us to interpret the experiments with only AMC-109 at the mica surface as small aggregates of AMC-109 molecules. MD simulation were employed to obtain a molecular picture of how such aggregates interact with a membrane surface, an information that is unavailable from our experiments (Fig. 4b). The specificity of the AMC-109 aggregates towards bacterial membranes was also predicted based on the MD simulations and then confirmed experimentally by HS-AFM measurements (Fig. 4c,d).

In the caption to SI video 4, we specify that the adsorption of AMC-109 on the membrane renders the membrane thinner, in accordance with experimental measurements. As the composition of the membrane in the simulations does not reach the complexity of the extracts from *S. aureus*, we did not want to compare the two except for the surface effects related to the negatively charged lipids, a phenomenon that is of a general nature. However, following the suggestion of the reviewer on the lipid order we now added analysis results that show that the simulated membrane thins, expands and the order of its lipid tails decreases (Fig. S10).

It seems that HS-AFM is capable of imaging all bilayer structures in this report. Is there any reason to use traditional AFM in this report (e.g. Fig. 1)?

Traditional AFM was used to observe the changes at a bigger length scale. Fig. 1 shows 3x3 μm view of the *S. aureus* lipid membrane. It allowed us to evaluate the average membrane expansion on a bigger area.

Reviewer #2 (Remarks to the Author):

Melcrova and colleagues show in this work the potential mechanism of action of the antibacterial peptidomimetic compound AMC-109. This peptide is already being tested for antibacterial activity against skin infections and nasal colonization but surprisingly, the mechanism of action has not been elucidated yet. In this work, they suggest that AMC-109 causes an alteration of the lateral membrane organization that reduces bacterial viability. The authors claimed that this is specific, or more selective, to bacterial membranes than eukaryotic membranes. Overall, this is a very promising line of research and the potential antibacterial activity of AMC-109 may develop into a new class of antibiotic. It may also open a new venue to selectively kill antibiotic-resistant bacteria by altering bacterial membrane organization thus affecting many cellular processes at once.

There are three important aspects of the mechanism of action of AMC-109 that are not sufficiently addressed in this work and need to be developed further to clarify how AMC-109 works. One is related to the selectivity of AMC-109 against bacterial membranes against eukaryotic membranes; the second aspect is related to the activity against entire bacteria rather than purified membranes or artificial membranes. The third aspect is related to how fast would the resistance against AMC-109 occur in bacteria that are exposed to the antibiotic. This third aspect is key in the microbiology field to elucidate/confirm a potential mechanism of action of any given antibiotic.

1) How selective is AMC-109 against bacterial membranes? The authors claimed that AMC-109 is more selective against negatively-charged membranes of bacteria than neutral eukaryotic membranes. This is an interesting point but it needs to be validated *in vitro* and *in vivo*. For instance, bacteria have a cell wall that acts as a barrier against many molecules. It may require a concentration higher than expected to penetrate the cell wall and probably at this concentration is more toxic to exposed eukaryotic membranes. In addition, even if the compound does not affect the membrane organization of eukaryotic membranes, it would be important to know whether it integrates in eukaryotic membranes and forms aggregates. This is important because the accumulation of aggregates in cellular is known to cause severe pathologies.

We agree this is an important point and we should have stressed this more. Selectivity for bacteria over human cells was published in Isaksson et al. *J. Medicinal Chemistry*, **16**(54), 5786–5795 (2011) (<https://pubs.acs.org/doi/10.1021/jm200450h>). Table 1, peptide 1 in this publication shows ~100-fold difference in MIC against *S. aureus* (<2 µg/ml) versus EC50 (erythrocyte toxicity; 175 µg/ml). Note that while the MIC is measured in a rich medium (MH-broth), the erythrocyte toxicity is measured in a dilute buffer to be able to observe the EC50 value.

Another study Saravolatz et al. *Antimicrobial Agents and Chemotherapy* **56**(8), 4478–4482 (2012) (<https://journals.asm.org/doi/10.1128/AAC.00194-12>) reports activities of AMC-109 (LTX-109 in the study) against a large number of resistant *S. aureus* clinical isolates including MRSA (96 strains), VISA (33 strains), and VRSA (13 strains). All show similar MIC values of 2–4 µg/ml. The manuscript is now updated to include more information on the *in vivo* studies.

2) The activity against entire bacteria rather than purified membranes or artificial membranes. Most of the work involved in the characterization of AMC-109 has been performed using purified membranes or artificial membranes. To be clear, this referee does not criticize the experiments and modelling presented in this work but it should be combined with experiments using living bacteria. Again as an example, bacteria have many ways to maintain membrane homeostasis and probably they have potential mechanisms to counteract the effect of AMC-109 that are not known yet. It is important to validate the *in vitro* results using purified membranes in living cells.

We also agree here that this is very important. In our study we observe changes on both *S. aureus* lipid membranes and artificial phospholipid membranes in a range of 1–4 µg/ml AMC-109, which is

the same as the MIC concentrations reported for living *S. aureus* bacteria. In particular, Isaksson et al. *J. Medicinal Chemistry*, **16**(54), 5786–5795 (2011) reports a MIC value against *S. aureus* of <2 µg/ml. We now rephrased and expanded the info on this point in the manuscript to make this clearer.

3) Evolution of resistance against AMC-109. The authors claimed that it might take a while before resistance is generated in bacteria. This is possible and would be fantastic but this needs to be tested. It is actually an important step for a microbiologist to elucidate the resistant mechanism (or at least to identify mutations that confer resistance) because from these mutations one can validate the mechanism of action of the compound. Possibly the resistance mechanism is based on preventing the contact of AMC-109 with the bacterial membrane by increasing the thickness of the cell wall, similar to the intermediate-resistance mechanism to vancomycin of *S. aureus* clinical strains (VISA). The authors at least should try how effective is AMC-109 against a VISA strain (a model VISA strain such as Mu50) because this may support the notion that AMC-109 needs to pass the cell wall to be active and thus is consistent with the hypothesis that it attacks lateral membrane organization.

The compound was previously tested against multiple resistant strains. In particular, Saravolatz et al. *Antimicrobial Agents and Chemotherapy* **56**(8), 4478–4482 (2012) reports activities of AMC-109 (LTX-109 in the study) against a large number of resistant *S. aureus* clinical isolates including MRSA, VISA, and VRSA. For all these studies similar MIC values of 2–4 µg/ml were found. AMC-109, hence, is capable of transferring through a thicker membrane of resistant *S. aureus* strains. We agree that this is a valid question, so we added an explanation of this point to the manuscript.

Indeed, we do not state that resistance will never occur. We raise an argument that resistance is not likely to develop quickly. We back up our argument with the similarities with small disinfectants that are in use for many years without developing resistance, and the fact that AMC-109 is able to transport through thicker cell wall of already resistant strains as was pointed out by this reviewer.

Minor comments

Why the authors focus on membrane damage on the first time? There is no initial indication that AMC-109 caused membrane damage. How they concur that the membrane is the target?

AMC-109 was designed as a minimal size mimic of a membrane-active antimicrobial peptide. A membrane-active peptide needs to be at least 4 amino acids long, has to have at least 2 cationic residues to ensure attractivity to the negatively charged bacteria, and sufficiently large hydrophobic part to ensure its insertion into the membrane. It was proposed by the authors of the AMC-109 design that its target is the membrane. The specifics of the membrane-antibiotic interactions were, however, unknown and became the core topic of our study. We briefly discuss the peptidomimetic design in the introduction (including references to literature [7,11,12]). However, as this has been published before, we prefer not to expand on this more and to focus on the core topic as mentioned above.

The authors claimed that a single membrane leaflet and its effects on the lateral organization in fact resemble the activity of small disinfectant molecules like ethanol, benzalkonium chloride (BAK) or benzyl alcohol. These are used as disinfectants but they cause damage in eukaryotic cells. a comparative study of damage in bacterial and eukaryotic cells is needed in this work.

Several studies on cytotoxicity of mentioned disinfectants have been performed. Groothuis et al. *Chemical Research in Toxicology* **32**(6), 1103–1114 (2019) performed systematic evaluation of BAK cytotoxicity. Table 3 in this study shows the cytotoxicity (EC50 value) of BAK with varying chain lengths. 31 µM for BAK with 10 carbons equals 8.8 µg/ml. For longer chain lengths, the (EC50 value) varies between 0.5 to 1.3 µM, corresponding to 0.2–0.7 µg/ml. The MIC values against *S. aureus*

bacteria for BAK are in the same range (2–4 µg/ml for MSSA and 2–16 µg/ml for MRSA) (reported by Raggi et al. *Clinical Microbiology* **2**(6) (2012), <http://dx.doi.org/10.4172/2327-5073.1000121>). Cytotoxic activity of ethanol is also well documented. Wu and Cederbaum *The Journal of Biological Chemistry*, **271**(39) (1996), <http://dx.doi.org/10.1074/jbc.271.39.23914>, reports ethanol toxicity on human HepG2 cells already at 10 mM ethanol, which equals 0.06 vol%. Tests of the ethanol toxicity on neural cells showed cytotoxicity using 100 mM ethanol (Chen and Sulik *Alcoholism: Clinical and Experimental Research*, **20**(6) (1996), <https://onlinelibrary.wiley.com/doi/10.1111/j.1530-0277.1996.tb01948.x>). Ethanol antimicrobial effects were, however, shown only for 5 vol% and higher concentration (Oh and Marshall *International Journal of Food Microbiology*, **20**(4) (1993), <https://linkinghub.elsevier.com/retrieve/pii/016816059390168G>).

The BAK compound, ethanol, and other disinfectants, hence, have no specificity towards bacteria over human cells, and as such cause a lot of damage to eukaryotic cells in concentrations that kill bacteria. We now include a reference to the Groothuis *et al.* 2019, Wu and Cederbaum 1996, Chen and Sulik 1996, and Oh and Marshall 1993 papers so that the interested reader knows where to find more information.

The authors claimed that AMC-109 is not a pore former using a pore-former molecule melittin as control. Totally honest, I have never heard of this pore former. What are the differences between this pore former and other classical pore formers such as nisin or colistin?

Melittin is a single α -helix antimicrobial peptide from honey bee venom. Melittin was isolated from honey bee venom in mid-twentieth century and has been studied for instance for its membrane activity. Melittin pores in phospholipid membranes were previously directly observed using AFM (Juhaniwicz, J. & Sek, S. *Electrochim Acta* **197**, 336–343 (2016) <http://dx.doi.org/10.1016/j.electacta.2015.11.134>; Giménez, D., Sánchez-Muñoz, O. L. & Salgado, J. *Langmuir* **31**, 3146–3158 (2015) <https://pubs.acs.org/doi/10.1021/la504293q>). The internal structure of the pores was modelled by multiple MD simulation studies (Miyazaki et al. *BBA Biomembranes* **1861**(7), 1409–1419 (2019), <https://doi.org/10.1016/j.bbamem.2019.03.002>; Lyu et al. *Journal of Chemical Physics* **146**(15) (2017), <http://dx.doi.org/10.1063/1.4979613>), and experimentally confirmed by oriented circular dichroism in combination with neutron scattering (Yang et al. *Biophysical Journal* **81**(3), 1475–1485 (2001); [http://dx.doi.org/10.1016/S0006-3495\(01\)75802-X](http://dx.doi.org/10.1016/S0006-3495(01)75802-X)). Similarly to AMC-109, melittin does not have any specific binding molecule in the membranes of bacteria. AMC-109 was designed to mimic such unspecific antimicrobial peptides as melittin, so it was chosen for the comparative study. The mentioned references are added into the manuscript.

Nisin on the contrary has a specific binder in the membranes of bacteria, which is lipid II. Nisin is recruited to bacterial membranes by binding to lipid II, which is followed by insertion of nisin into the membrane and pore creation.

Colistin belongs to polymyxin antibiotics, which are used to treat only Gram-negative bacterial infections as they are active at the outer Gram-negative membrane. Colistin selectively binds lipopolysaccharide that is not present in the membrane of Gram-positive *S. aureus* discussed in our study.

To summarize, in our choice of pore former we wanted to avoid the need for any specific binding partner in the bacterial membrane. Both nisin and colistin, however, need specific molecules (lipid II or lipopolysaccharides) for their poration activity.

Reviewer #3 (Remarks to the Author):

The authors provide data that sheds light on the functional mechanism of the known peptide AMC-109. Although the experimental work provided is relatively sound, the article lacks the level of novelty expected for Nature Communications.

Specific comments:

- Peptide AMC-109 has been described previously, significantly decreasing the novelty of this study.

We don't think that the use of a very relevant peptide in clinical development is decreasing the novelty (or importance) of the study. The aim of the study is not to develop new antibiotic compound, but to shed light into its mode of activity that was not described before. We believe that our discovery of a new antibiotic target in the bacterial membranes (their lateral organization) is of significant importance and a clear novelty.

- AMC-109 should be referred to as a peptide, not a peptidomimetic, since the backbone is composed of amide bonds and the alpha carbons as in a regular peptide. Non-canonical residues do not necessarily make it a peptidomimetic, and the capped N-terminal does not warrant this classification either in my view. Examples of peptidomimetics would be peptoids, peptides composed of beta-amino acids, etc. The arginine residues are not arginine-like residues, they are arginine residues.

The use of the word peptidomimetic means a molecule that mimics a peptide. In the present case, AMC-109 has been designed as a smallest possible mimetic of a natural antimicrobial peptide, hence, we use the word peptidomimetic. We agree that AMC-109 has regular peptide bonds. However, the peptide contains a modified tryptophan residue. The presence of three tert-butyl groups in the 2-, 5- and 7-positions on the indole ring makes the side chain of this amino acid 2.5 times larger and more lipophilic than the indole of tryptophan. This is a key element of AMC-109, as it allows the peptide to obtain high activity with only three amino acids. Moreover, the C-terminus is also capped. This fact together with the use of unnatural amino acids in our opinion justifies the use of the word "peptidomimetic". Therefore we prefer to stick with this terminology. We have added a clarification to the introduction.

We changed the word "arginine-like" into "arginine" in the manuscript.

- All concentrations should be indicated in μM , which make AMC-109 very average in terms of bioactivity.

We do not agree that it would be more appropriate to use μM concentrations in case of AMC-109, neither that it is average in terms of bioactivity. μM concentration refers to a certain number of molecules, which is relevant when one has a target with one point of occupancy (like an enzyme and its inhibitor). AMC-109 as our work shows, is working through a mass effect, hence an activity measurement in $\mu\text{g}/\text{ml}$ is not only warranted, it is also the most appropriate measurement.

Moreover, we see AMC-109 as more active than comparable alternatives considering that reported activity of the most active cationic antimicrobial peptides stops at around $2 \mu\text{g}/\text{ml}$. A comparative study of the antimicrobial activities of six different antimicrobial peptides against methicillin-resistant *S. aureus* (MRSA) was shown in Ciandrini et al. *Current Topics in Medicinal Chemistry* **18**(24), 2116–2126 (2018) (<https://www.eurekaselect.com/166508/article>), which states that "The lowest MIC value was observed for Pal-KGK-NH₂ ($1 \mu\text{g}/\text{ml}$), followed by Temporin A ($4-16 \mu\text{g}/\text{ml}$), CA(1–7)M(2–9)NH₂ ($8-16 \mu\text{g}/\text{ml}$) and Citropin 1.1 ($16-64 \mu\text{g}/\text{ml}$), while higher MICs were evidenced for LL-37,

Pexiganan and Pal-KKKK-NH₂ (> 128 µg/ml).” In contrast, AMC-109 has been shown active in 2–4 µg/ml for multitude of resistant *S. aureus* strains (see response to Reviewer #2, point 1). This shows that it is not “average”.

- The authors should discuss whether a less hindered Trp derivative would be beneficial or not for the proposed mechanism of action. The tert-butyl groups would only make it difficult for the peptides to aggregate and would delay the kinetics, which seems to be important for mechanism of action. The Trp analog would make a good control.

The Trp version of AMC-109 is well described in literature in Svenson et al. *Biochemistry* **47**(12), 3777–3788 (2008) (<https://pubs.acs.org/doi/10.1021/bi7019904>). The mentioned study shows different AMC-109 derivatives, Trp derivative being labeled CAP 16 (see Fig. 2 in that study). Table 3 clearly shows that Trp derivative CAP 16 displays very low activity towards *S. aureus* with the MIC of 145 µM (2 µg/ml of AMC-109 equals 2.5 µM) showing that the less hindered Trp derivative is not beneficial for the activity of AMC-109. We now discuss this in the manuscript.

- According to the aggregate size distribution results, please discuss if this mechanism of action is expected for an aggregate which is almost the size of a lipid bilayer (3-4 nm in length).

We do not understand this question. Both AFM as well as MD shows aggregates of 3-4 nm. Furthermore, our MD simulation clearly shows that these aggregates of 3–4 nm attach to the membrane and gradually dissolve the AMC-109 molecules in between the lipids (Fig. 4b). This mechanism is clearly suggested exactly for the size of aggregates that Reviewer #3 mentions.

- Why are the MD simulations done in membranes that are different from those found in the lipidomics study? POPC/POPG (MD) vs CL/PGLysyl-PG (lipidomics). The same comment applies for the calcein leakage studies (POPG/POPC/POPE).

The lipidomics study reveals a complex mixture of lipids, majority of which are glycerophospholipids and glyceroglycolipids. Our lipidomics MS analysis reveals at least 101 glycerophospholipid masses (distributed over 3 lipid headgroup classes: PG, CL, and Lysyl-PG,) and 63 glyceroglycolipid (denoted by glycolipid in the manuscript, as is often done in literature) masses (distributed over 2 lipid headgroup classes: 1-hex-DAG and 2-hex-DAG). Each detected mass belongs to one of those classes (i.e. contains that headgroup) and has a certain amount of carbons and double bonds in its radyl groups. Each of these masses with a certain headgroup, radyl-carbon number, and double bonds; has a range of possible regioisomers pertaining to double bond location, iso-/anteiso- isomerism, and various radyl tail combinations possible to make up that carbon/double bond count. Overall, this results in at least hundreds of theoretically possible individual lipid species. We couldn't resolve the relative ratio of those glycerophospholipids with the second biggest lipid group, the glyceroglycolipids. Hence, we analyze these two lipid groups separately and show the results in Table 1. Table 1 only shows the relative densitometric response of the three major glycerophospholipid classes (and the two glyceroglycolipid classes) in the *S. aureus* membrane extract on a TLC plate with molybdenum blue staining (Fig. S1a and c) and α-naphtol staining (Fig. S1d) respectively. Each spot on the TLC corresponds to a lipid class and all radyl tail configurations and isomers and such of a class are contained within the TLC spot, greatly simplifying quantification of lipid class proportions. It does not by far provide a full overview of the membrane complexity. Taken together, the huge diversity in lipids makes modelling the membrane of *S. aureus* in its full complexity not feasible, as (i) the lipid head/tail combination is not resolved, (ii) details about lipid leaflet asymmetry are lacking, and (iii) validated topologies for all lipid types are not available for the chosen force field. Therefore, we decided to use a prototypical model for a plasma membrane consisting of POPC/POPG with varying amounts of PG to mimic the difference between bacterial membranes (containing about 60% of anionic lipids) on the one hand, and eukaryotic membranes (<10% PG content) on the other.

We now update the manuscript with more details about the lipidomic analysis to avoid confusion. The table headers and caption have been updated to emphasize that the quantification was performed using TLC, and thus was done on a class-level.

In case of leakage assays, we opted for simple membrane composition for two reasons. First, encapsulation of calcein into POPG/POPC/POPE liposomes is a standard procedure that does not require long fine-tuning and could be compared well to literature. These liposomes are stable with no leakage of their own. Often with complex cellular extracts, leakage assays are challenging because the liposomes are slightly leaking already before the antibiotic addition. Second, to get enough fluorescence signal at leakage assays, we need to have a final 100 mg/ml concentration of liposomes in a 1 ml volume. Extraction and characterisation of *S. aureus* lipid membranes is a lengthy and expensive procedure. Using these extracts for leakage assays would use a majority of the extracted membranes. In contrast, for AFM and HS-AFM studies we prepare 250 μ l stocks of only 0.3 mg/ml liposomes that can be used for a full week of experiments. Final amounts used for a single AFM experiment is 3-10 μ l of 0.03 mg/ml liposomes.

- How translatable are the results obtained in a 2D lipid coating experiment compared to the membrane of a microorganism, considering tension and other physical parameters that will most likely influence how the lipid bilayer will respond to the peptide or aggregate upon contact?

The reported MIC concentrations in various *S. aureus* strains are the same as the concentration when we observe changes on the supported membranes (see response to point 1 of Reviewer #2). The bilayer in the living organism cannot expand to sides as on our support. The membrane could become rough or form protrusions as a result of the build-up of tension. However, our system remains a model system indeed. Follow up studies observing the effects in living microorganisms would certainly be useful.

- Why was melittin used as a comparative control when it has a completely different mechanism of action (pore-forming)?

AMC-109 was designed as a mimic of a membrane-active antimicrobial peptide. Isaksson et al. *J. Medicinal Chemistry*, **16**(54), 5786–5795 (2011) postulated that its target is the membrane with a pore formation being its likely mechanism. All our data show there is no pore formation activity present. The comparison to the pore-former melittin makes this conclusion solid. However, we agree that in the section “AMC-109 does not form membrane pores” we do not introduce our motivation for performing the melittin experiments properly. We now changed this to explain our intention better.

- Please add positive controls for the first dissolution of lateral domains experiments.

We do not know what positive controls are meant. We report a new mechanism of antimicrobial activity, which is dissolution of lateral domains. We are not aware of any other studies and compounds that would serve as a positive control for the observation of lateral domains dissolution.

- More recent references should be added.

We refer to the most relevant references for each aspect of the study. Some of the references are of an older date, but contain relevant information for the study in hand. We added references to aspects that were pointed out by the Reviewers:

- Svenson, J. *et al.* Antimicrobial Peptides with Stability toward Tryptic Degradation. *Biochemistry* **47**, 3777–3788 (2008).
- Guha, S., Ghimire, J., Wu, E. & Wimley, W. C. Mechanistic Landscape of Membrane-Permeabilizing Peptides. *Chem Rev* **119**, 6040–6085 (2019).
- Juhaniewicz, J. & Sek, S. *Electrochim Acta* **197**, 336–343 (2016) (<http://dx.doi.org/10.1016/j.electacta.2015.11.134>)
- Giménez, D., Sánchez-Muñoz, O. L. & Salgado, J. *Langmuir* **31**, 3146–3158 (2015) (<https://pubs.acs.org/doi/10.1021/la504293q>)
- Groothuis, F. A. *et al.* Influence of in Vitro Assay Setup on the Apparent Cytotoxic Potency of Benzalkonium Chlorides. *Chem Res Toxicol* **32**, 1103–1114 (2019).
- Wu and Cederbaum *The Journal of Biological Chemistry*, **271**(39) (1996) (<http://dx.doi.org/10.1074/jbc.271.39.23914>)
- Chen and Sulik *Alcoholism: Clinical and Experimental Research*, **20**(6) (1996) (<https://onlinelibrary.wiley.com/doi/10.1111/j.1530-0277.1996.tb01948.x>)
- Oh and Marshall *International Journal of Food Microbiology*, **20**(4) (1993) (<https://linkinghub.elsevier.com/retrieve/pii/016816059390168G>).
- Miyazaki *et al.* *BBA Biomembranes* **1861**(7), 1409–1419 (2019) (<https://doi.org/10.1016/j.bbamem.2019.03.002>)
- Lyu *et al.* *Journal of Chemical Physics* **146**(15) (2017) (<http://dx.doi.org/10.1063/1.4979613>)
- Yang *et al.* *Biophysical Journal* **81**(3), 1475–1485 (2001) ([http://dx.doi.org/10.1016/S0006-3495\(01\)75802-X](http://dx.doi.org/10.1016/S0006-3495(01)75802-X))
- added to SI: Saavedra *et al.* *Langmuir* **36**, 5709–5716 (2020) (<https://pubs.acs.org/doi/pdf/10.1021/acs.langmuir.0c00247>)
- added to SI: Majewska *et al.* *BBA Biomembranes* **1862**(9) (2020) (<https://doi.org/10.1016/j.bbamem.2020.183347>)
- added to SI: Buchoux, S. *Bioinformatics* **33**, 133–134 (2017)
- added to SI: Melcr *et al.* *J Chem Theory Comput* **16**, 738–748 (2020)

We also added the following references to justify our choice of AFM and HS-AFM for an antimicrobial mechanism study:

- Manioglu, S., Modaresi, S.M., Ritzmann, N. *et al.* *Nat Commun* **13**, 6195 (2022). (<https://doi.org/10.1038/s41467-022-33838-0>)
- Zuttion, F., Colom, A., Matile, S. *et al.* *Nat Commun* **11**, 6312 (2020). (<https://doi.org/10.1038/s41467-020-19710-z>)

Reviewer #4 (Remarks to the Author):

This is a nice, mainly AFM study on the antimicrobial peptidomimetic tripeptide AMC-109, developed more than a decade ago and extensively tested in clinical trials since. The authors do lipidomics to come up with a synthetic model of the membrane of *S.aureus* (w/o peptidoglycan) and use that in the AFM experiments.

The AFM pictures look beautiful and it is convincingly demonstrated experimentally that AMC-109 adsorbs into these membranes, expands them laterally, dissolving lipid rafts and thinning the membrane overall.

That this is the way *S.aureus* is killed is conjectured from two controls: The same AFM methods are

used on the toxic honey bee peptide melittin, which – in contrast – shows poration as the mechanism. This is also corroborated by a calcein leakage assay that shows poration for melittin and not for AMC-109. The second comparison is to the small disinfectant molecule benzalkonium chloride (BAK), which shows results very similar to AMC-109. So the mechanism is claimed to be a change in membrane lateral organization and lipid order, and dissolving membrane domains. It is hypothesized that the bacterium cannot survive if its functional membrane microdomains (FMMs) are dissolved, although no direct evidence is gathered for this here as the experiments are purely on synthetic bilayer constructs, not organisms.

A bit curious is the addition of coarse grain (CG) MD simulations as a way to visualize what is going on. Firstly, I'd like to point out that matching the results of experimental assays and MD simulations is a very challenging task, so the authors should be commended for attempting it. However, here I have some observations that appear incongruous:

- The whole manuscript is focused on lipid domains and their dissolution, as well as the overall membrane thinning due to incorporation of AMC-109. Yet, the CG MD simulations do not show any of this. No lipid domains are visible. And no membrane thinning is demonstrated.

The focus of the paper is the mode of action of AMC-109, which we reveal is membrane reorganisation. We agree that the MD simulations do not capture all effects we see experimentally, but that was also not to be expected due to the large difference in spatial and temporal scales between both approaches. Nevertheless, the thinning effect was captured, and we agree with the referee that this warrants demonstration. In accordance with the experiments, the effect of AMC-109 on membrane thinning is visible in SI video 4, as we now specify in the caption, and in the new Fig. S10.

No lipid domains are visible as we used simple model compositions that do not form domains in MD simulations. The choice was guided by the aim to make a systematic assay that can be easily interpreted. This way, we have arrived at a conclusion that sufficient amount of negative charge is required for the AMC aggregates to interact with the membrane surface. Simulations of complex membranes that would include lipid domains are indeed very interesting, but will require a huge amount of work, which is clearly beyond the scope of our work (see also reply to Reviewer #3 above).

- The CG MD uses POPC/POPG bilayers. Significant lipidomics was done to arrive at a simple model of the membrane of *S.aureus*. Why did the CG simulations not simply employ those lipid compositions?

Indeed this is an important point that was not explained well in the original manuscript. We repeat our answer to Referee #3, who raised the same question: The lipidomics study reveals a complex mixture of lipids, majority of which are glycerophospholipids and glyceroglycolipids (in literature for simplicity often glycolipid is written instead of glyceroglycolipid, in accordance in the manuscript we simply write glycolipid). Our lipidomics MS analysis reveals at least 101 glycerophospholipid masses (distributed over 3 lipid headgroup classes: PG, CL, and Lysyl-PG,) and 63 glyceroglycolipid masses (distributed over 2 lipid headgroup classes: 1-hex-DAG and 2-hex-DAG). Each detected mass belongs to one of those classes (i.e. contains that headgroup) and has a certain amount of carbons and double bonds in its radyl groups. Each of these masses with a certain headgroup, radyl-carbon number, and double bonds; has a range of possible regioisomers pertaining to double bond location, iso-/anteiso- isomerism, and various radyl tail combinations possible to make up that carbon/double bond count. Overall, this results in at least hundreds of theoretically possible individual lipid species. We couldn't resolve the relative ratio of those glycerophospholipids with the second biggest lipid group, the glyceroglycolipids. Hence, we analyse these two lipid groups separately and show the results in Table 1. Table 1 only shows the relative densitometric response of the three major glycerophospholipid classes (and the two glyceroglycolipid classes) in the *S. aureus* membrane

extract on a TLC plate with molybdenum blue staining (Fig. S1a and c) and α -naphthol staining (Fig. S1d) respectively. Each spot on the TLC corresponds to a lipid class and all radical tail configurations and isomers of a class are contained within the TLC spot, greatly simplifying quantification of lipid class proportions. It does not by far provide a full overview of the membrane complexity. Taken together, the huge diversity in lipids makes modelling the membrane of *S. aureus* in its full complexity not feasible, as (i) the lipid head/tail combination is not resolved, (ii) details about lipid leaflet asymmetry are lacking, and (iii) validated topologies for all lipid types are not available for the chosen force field. Therefore, we decided to use a prototypical model for a plasma membrane consisting of POPC/POPG with varying amounts of PG to mimic the difference between bacterial membranes (containing about 60% of anionic lipids) on the one hand, and eukaryotic membranes (<10% PG content) on the other.

- CG MD predicts that AMC-109 aggregates are adsorbed and dissolved after 300 ns - 2 μ s (Figure 4a). After that, individual AMC-109 molecules are laterally diffusing among the lipids in the upper bilayer leaflet and a steady state appears reached. Yet the experiments show timescales of 4 min for domain dissolution and 15 min for bilayer thinning. That is an 8 order of magnitude difference. Could the authors give an explanation?

This is a valid concern. Comparing time scales between macroscopic experiments and simulations performed at the nanoscale is notoriously difficult. There are a number of reasons why the time scales are not comparable. The simulated system uses a simple model membrane composition that equilibrates quickly. This is not the case in experiments. The experimental setup uses supported lipid bilayers, which are known for slower dynamics compared to free standing lipid bilayers used in the simulations. Reaching equilibrium dissolution of AMC in a fluid bilayer has a different time scale in this situation. In addition, the kinetics of AMC adsorption to the membrane is likely very different between the experimental and simulation setup — in the latter case the peptides are already present close to the membranes from the start, whereas in the experiment a large aqueous volume is present with a much lower effective peptide concentration near the membrane. As the difference in time scales between MD and experiments occurs more often and especially as we can't say whether above explanations account for the 8 orders of magnitude difference, or only for — let's say - 6 or 5 orders of magnitude and we still miss out on something in our explanation, we decided not to speculate about it in the manuscript.

- A nice result is the demonstration that AMC-109 aggregates in solution, and that the AFM measurements seem to see and corroborate the size of these aggregates (Figure 3).

We thank the reviewer for the comment.

- When these aggregates are then let loose to adsorb on the membrane, they are seen selective in binding for PG and not PC. CG force fields have very smooth energy landscapes that are ideally suited to study aggregation, adsorption or other phenomena with large, homogeneous phases, such as lipid membranes. The effect seen here (and shown many times before with other aggregates on membranes) is the rapid minimizing of the free energy due to charge compensation (i.e. positively charged CG beads get close to negatively charged CG beads). As the authors show, there is a direct correlation of AMC-109 adsorption with the concentration of anionic PG lipids, so that one AMC-109 (charge +3) adsorbs per 3 PG (charge -1) until the membrane is neutral and then stops. Why? What happens to the remaining AMC-109 molecules? Do they stay in solution? In this case, a neutral bilayer would not be the preferred free energy minimum location of AMC-109, but rather it prefers the a hydrophobic self-aggregate instead of the hydrophobic membrane core as a way to bury its hydrophobic parts. So it stays in solution as hydrophobic aggregate when membranes are neutral and only loves anionic membranes.

Yes, this is a nice summary of our conclusions from the MD simulations. We believe that the saturation of AMC adsorption indeed happens approximately at the level when the charges on the membrane are compensated. As the resulting membrane is approximately neutral overall, the adsorption is suppressed and the rest of AMC molecules remains in solution as small micellar aggregates. We rephrased now these parts in the manuscript to make this clearer.

- This seemingly explains the lack of insertion in the PC bilayer. However, the reason is then not a kinetic one, as the authors speculate (quote: “Second, self assembly of AMC-109 creates an energetic barrier preventing interaction of AMC-109 with neutral membranes”). Given the smoothness of CG force fields, which lack hindering hydrogen bonds slowing kinetics, this does not sound too convincing. It could be checked by just putting AMC-109 molecules into the PC bilayer from the start and seeing if they stay there. This would make the bilayer positively charged, but the counterions compensate. The current results are puzzling, because in the HF-AFM measurements, AMC-109 clearly binds to pure POPC and increases its surface by 16% (Figure 4C), whereas in the CG MD, it does not bind at all. Even more puzzling: If the membrane is made positively charged (POPC/DOTAP), there is a significant 50% increase in surface area, indicating substantial AMC-109 adsorption. This seems to contradict the CG MD observations that insertion is merely proportional to anionic charge content due to 3:1 charge compensation. Clearly, AMC-109 also loves cationic membranes, and neutral ones a bit. The authors even mention this (quote: “AMC-109 in its monomeric form is highly hydrophobic so it tends to interact with all membranes including those of mammalian cells”).

We did not intend to express that the reason for the lack of insertion to neutral membranes is a kinetic barrier. Our wording in the mentioned sentence is probably misleading. We wanted to express exactly the point made by the reviewer that “the neutral membrane is not a preferred free energy minimum location of AMC-109, but rather it prefers the a hydrophobic self-aggregate instead of the hydrophobic membrane core as a way to bury its hydrophobic parts”. We changed the mentioned sentence in the manuscript to avoid confusion.

Concerning the comparison between experiments and simulations regarding the affinity of the peptides for neutral (or cationic) membranes, we want to point out that the equilibrium between embedding in the membrane versus forming micelles in solution is strongly dependent on overall peptide concentration. The concentration of AMC-109 in simulations is much higher than the concentration used in experiments, simply because the aqueous volume that can be taken into account in the simulations is limited. A higher concentration leads to a stabilization of the micellar phase. Note, however, that there is a small, non-zero, amount of interaction between AMC-109 and the neutral membrane also in the simulations. In the rare event when an AMC-109 molecule leaves its micellar aggregate, we have observed that such molecules may enter the membrane and stay there until the end of the simulation. This indicates that monomers indeed prefer the hydrophobic interior of the neutral membrane rather than being freely in water. The fact that monomers prefer insertion over free water solution was also previously shown in all-atom MD simulations and NMR studies in Isaksson et al. *Journal of Medicinal Chemistry* **54**(16), 5786–5795 (2011) (<https://pubs.acs.org/doi/10.1021/jm200450h>), a publication to which we refer to in the manuscript. The suggestion to put AMC-109 molecules into the PC bilayer from the start and seeing if they stay there is interesting. Looking at our simulations made us realize we already have this data, but that we did not analyse it yet. Our simulation setup starts with many AMC-109 monomers in the bulk water surrounding the membrane. In the initial phase of the simulation, when the aggregates form, a very small amount of monomers entered the PC bilayer instead. These monomers stayed in the membrane for the whole duration of the simulation. We stress this now out in the manuscript and in a new Fig. S9. Moreover, a short simulation with an atomistic model run (50 ns) of AMC-109 monomer inserted inside a POPE membrane was performed by Isaksson et al. 2011. The molecule was stable inside the membrane during that short time.

These differences in the HS-AFM results and CG MD are difficult to explain. I suspect that the mechanism derived from CG MD may be a bit too simplistic. If it is just a kinetic barrier, or just charge compensation, then any cationic peptide or peptidomimetic, many of which also form aggregates in solution, would be a good antimicrobial or anti-cancer drug, as those targets have anionic membranes. However, most cationic peptides are cytotoxic to eucaryotic cell membranes, which are not negatively charged.

We agree that the mechanistic picture only from MD simulations would be too simplistic in this case. Hence, we draw main conclusions from experiments and use simulation where appropriate to support and augment our findings. We do not recall any antimicrobial peptide that would form stable aggregates in solution that would form a minimum in the free energy of the peptide preventing it to enter neutral membranes. Antimicrobial peptides are in general bigger molecules than AMC-109 and aggregate into bigger structures like amyloid fibrils or unstructured peptide aggregates. These aggregates do not have all the hydrophobic parts of their molecules shielded from the surrounding water, hence, the membrane insertion is energetically favourable for them regardless the membrane charge. The charge of antimicrobial peptides makes them attractive to bacteria, increasing their antimicrobial activity, but the type of aggregates does not prevent cytotoxicity as we propose for AMC-109.

It would be nice if the authors made some more control simulations to address these discrepancies between HS-AFM and CG MD, and try to corroborate their theory that the lack of toxicity of AMC-109 is due to a kinetic barrier in unpacking the aggregates, not thermodynamic reasons.

We do not state that the reason for not entering the neutral membrane is a kinetic barrier. We only state that the aggregates are stable in water and in an energetic minimum. The reference supporting our hypothesis is also to a thermodynamic study. The kinetic barrier theory was probably misunderstood from our original wording mentioned by Reviewer #4 in one of their previous points. We agree that that was not clear, so we changed the corresponding sentence to avoid such confusion. We also agree that a control simulation would be very useful. As written above, we now added simulation results of monomers in a PC bilayer looking whether they are stably incorporated. Indeed these monomers stayed in the membrane for the whole duration of the simulation. We discuss this now in the manuscript and added a new figure (Fig. S9) showing the results.

Minor points:

- Fig. 3b: The picture of AMC-109 in CG representation looks really strange. Maybe a stick-model is not suited. It would be better to show how this molecule really looks in CG representation, i.e. CG beads or ball-and-stick model? So it can be judged how much information is missing as compared to an all-atomistic model.

The amount of missing description is directly coming from the utilised model, Martini 3 (Marrink et al. *Wiley Interdiscip Rev Comput Mol Sci* 1–42 (2022)). The representation in Figure 3 in the manuscript is now enriched with a more common sphere representation, in which all interaction particles are represented as spheres. Mapping of the representation on the constituting (modified) amino acids is now apparent from the updated Fig. 3 that directly compares the representation with a chemical drawing. The colour code remains as: blue for arginine side chains, rest grey. We keep also the stick representation in Fig. 3 as it shows the connections between the interaction sites. The stick representation is further used in Fig. 4 and supporting videos.

- typo: l264 'Only a these very high concentrations' -> at

Corrected.

Reviewer #1 (Remarks to the Author):

The authors have addressed all my questions properly.

Reviewer #2 (Remarks to the Author):

The authors did not experimentally address any of my concerns. Therefore, my concerns still stand. The authors did not provide any experimental evidence of the mechanism that makes AMC-109 discriminate between bacterial and eukaryotic membranes. They cited several publications to confirm this, but this discriminating mechanism is intriguing and needs experimental clarification. A publication at this level requires a more substantial experimental basis to support the conclusions made in this work.

The authors did not include any *in vivo* experiments demonstrating that AMC-109 has antimicrobial properties against bacteria. All experiments are performed with artificial membranes. Similar to the above, the manuscript, at its current state is thin of experiments and needs more experimental evidence to support the authors' claims. Showing a battery of experiments using living bacteria and infection assays is critical to describe the function of the AMC-109 more clearly and convincingly.

The authors have not experimentally addressed my concerns on the evolution of resistance to AMC-109. This is a fundamental step in microbiology to identify the possible mechanisms of action of the antimicrobial and the route to bacterial resistance. It is not a challenging experiment to do; simply to grow bacteria with AMC-109 and sequence the genome of several resistant ones to identify the genes involved in the AMC-109 resistance and, thus the mechanism of action of AMC-109.

In general, the work still has the same issues and technical and conceptual inconsistencies as the initial version of the manuscript, because the authors have not experimentally addressed any of my concerns.

Reviewer #4 (Remarks to the Author):

The authors have answered all points raised convincingly. The changes made to the manuscript are acceptable. I have no further comments.

Rebuttal, round 2

Point-by-point response to reviewers' comments

Reviewer #1 (Remarks to the Author):

The authors have addressed all my questions properly.

We are glad that our replies and adjustments were sufficient.

Reviewer #2 (Remarks to the Author):

The authors did not experimentally address any of my concerns. Therefore, my concerns still stand. The authors did not provide any experimental evidence of the mechanism that makes AMC-109 discriminate between bacterial and eukaryotic membranes. They cited several publications to confirm this, but this discriminating mechanism is intriguing and needs experimental clarification. A publication at this level requires a more substantial experimental basis to support the conclusions made in this work.

We are sorry that Reviewer #2 was not satisfied with our first answer. Our study describes molecular mechanism of activity of AMC-109. The higher antimicrobial activity over cytotoxicity was already demonstrated in multiple studies so such experiments would be only repetition. Our study does not focus on whether AMC-109 is potent and not toxic antibiotic (this is already known), but on a molecular basis of its activity. We clearly demonstrate that AMC-109 acts upon bacterial membrane dissolving lipid nanodomains. We further demonstrate that AMC-109 forms aggregates in solution that have clear preference for negatively charged membranes (of bacteria) over neutral membranes (of eukaryotes).

The authors did not include any in vivo experiments demonstrating that AMC-109 has antimicrobial properties against bacteria. All experiments are performed with artificial membranes. Similar to the above, the manuscript, at its current state is thin of experiments and needs more experimental evidence to support the authors' claims. Showing a battery of experiments using living bacteria and infection assays is critical to describe the function of the AMC-109 more clearly and convincingly.

The publications we cite clearly prove that AMC-109 has much higher antimicrobial activity than cytotoxicity. We refer to these in vivo studies within the manuscript: Ref. 15 describes a pilot clinical Phase 2a experiment (i.e., on humans); Ref. 16 describes an infected wound treatment model in mice. Further to these two there is a recent publication where AMC-109 is tested in a rat burn model (www.sciencedirect.com/science/article/pii/S2213716521002885).

Infection assays (although useful for further investigations) cannot give molecular level (as MD simulations) or nanometer scale (as AFM) insight into the antibiotic-bacteria interaction, which is the core of our study.

The authors have not experimentally addressed my concerns on the evolution of resistance to AMC-109. This is a fundamental step in microbiology to identify the possible mechanisms of action of the antimicrobial and the route to bacterial resistance. It is not a challenging

experiment to do; simply to grow bacteria with AMC-109 and sequence the genome of several resistant ones to identify the genes involved in the AMC-109 resistance and, thus the mechanism of action of AMC-109.

AMC-109 has been studied and developed already for several years. This development has been lead by one of the co-authors of this study John S. M. Svendsen, who has been looking for years for spontaneous resistant strains and never found any. Furthermore, multiple pass experiments have all failed to yield resistant strains. The route suggested by Reviewer #2 is, hence, not feasible as it requires access to resistant strains. That said, we cannot exclude that there will never occur any resistant strain in future, however, it can be expected as a rare event.

In general, the work still has the same issues and technical and conceptual inconsistencies as the initial version of the manuscript, because the authors have not experimentally addressed any of my concerns.

Reviewer #4 (Remarks to the Author):

The authors have answered all points raised convincingly. The changes made to the manuscript are acceptable. I have no further comments.

We are glad that our replies and adjustments were sufficient.